# Molecular and clinical analyses of *PHF6* mutant myeloid neoplasia provide their pathogenesis and therapeutic targeting

Yasuo Kubota [1], Xiaorong Gu[1], Laila Terkawi[1], Juraj Bodo[2], Bartlomiej P. Przychodzen[1], Hussein Awada [1], Nakisha Williams[1], Carmelo Gurnari[1,3], Naomi Kawashima [1], Mai Aly[1], Arda Durmaz[1], Minako Mori [1], Ben Ponvilawan [1], Tariq Kewan [1], Waled Bahaj[1], Manja Meggendorfer [4], Babal K. Jha [1,5], Valeria Visconte[1], Heesun J. Rogers [2], Torsten Haferlach [4] & Jaroslaw P. Maciejewski [1] ✉

*PHF6* mutations (*PHF6*[MT]) are identified in various myeloid neoplasms (MN). However, little is known about the precise function and consequences of *PHF6* in MN. Here we show three main findings in our comprehensive genomic and proteomic study. Firstly, we show a different pattern of genes correlating with *PHF6*[MT] in male and female cases. When analyzing male and female cases separately, in only male cases, *RUNX1* and *U2AF1* are co-mutated with *PHF6*. In contrast, female cases reveal co-occurrence of *ASXL1* mutations and X-chromosome deletions with *PHF6*[MT]. Next, proteomics analysis reveals a direct interaction between PHF6 and RUNX1. Both proteins co-localize in active enhancer regions that define the context of lineage differentiation. Finally, we demonstrate a negative prognostic role of *PHF6*[MT], especially in association with *RUNX1*. The negative effects on survival are additive as *PHF6*[MT] cases with *RUNX1* mutations have worse outcomes when compared to cases carrying single mutation or wild-type.

Mutations in *PHF6* (*PHF6*[MT]) are responsible for congenital X-linked neurodevelopmental disorder Börjeson-Forssman-Lehmann syndrome (BFLS)[1]. Subsequently, acquired *PHF6*[MT] were detected in leukemias, chiefly T-cell acute lymphoblastic leukemia (T-ALL) but also in B-cell acute lymphoblastic leukemia (B-ALL) and various myeloid neoplasms (MN) including acute myeloid leukemia (AML) and myelodysplastic neoplasm (MDS)[2,3].

*PHF6* is highly expressed in brain/developing CNS as well as in hematopoietic subpopulations, potentially suggesting a dual role in both neurogenesis and hematopoiesis consistent with the neurologic presentation of germline mutations in BFLS and with leukemias in cases harboring somatic mutations[4,5]. Of note is that a male predominance has been observed in T-ALL and AML[2,3]. Although these results suggest that *PHF6* could escape from X chromosome inactivation, in female patients with T-ALL, *PHF6* showed monoallelic expression[2].

*PHF6*[MT] occurs early in T-ALL development as well as in mixed phenotype acute leukemia (MPAL), affects differentiation of blood cells, and facilitates the development of *NOTCH1* mutation-induced T-ALL[6–8]. PHF6 protein interacts with multiple nucleosome remodeling protein complexes including SWI/SNF or NuRD. Furthermore, PHF6 has been shown to bind to histones and regulate transcription of lineage-specific genes in T-ALL and B-ALL[7,9]. These results led to the identification of at least three general functions of *PHF6*: (i)

[1]Department of Translational Hematology and Oncology Research, Taussig Cancer Institute, Cleveland Clinic, Cleveland, OH, USA. [2]Department of Laboratory Medicine, Cleveland Clinic, Cleveland, OH, USA. [3]Department of Biomedicine and Prevention, University of Rome Tor Vergata, Rome, Italy. [4]MLL Munich Leukemia Laboratory, Munich, Germany. [5]Center for Immunotherapy and Precision Immuno-Oncology, Lerner Research Institute (LRI) Cleveland Clinic, Cleveland, OH, USA. ✉e-mail: maciejj@ccf.org

hematopoietic lineage differentiation, (ii) bone fide tumor suppressor, and/or: (iii) involvement in chromatin remodeling[7].

In MN, *PHF6* and consequences of its mutations have been less explored. *PHF6*[MT] was frequently found in secondary AML (sAML), MDS with excess blasts (EB), and blast crisis phase of chronic myeloid leukemia (CML)[3,10]. *PHF6*[MT] were more frequent in AML with trisomy 8, t(8;21) or complex karyotype, but mutually exclusive with *SF3B1*[3]. Detection of *PHF6*[MT] with germline *RUNX1* mutations (*RUNX1*[MT])[11] and those with somatic hits to *RUNX1*, *ASXL1*, or *U2AF1* indicates that *PHF6*[MT] are secondary events in myeloid ontogenesis[3]. This seems in contrast to T-ALL and MPAL, wherein *PHF6*[MT] constitutes early events[3,10]. In terms of clinical outcomes, *PHF6*[MT] may be an adverse event: e.g., in intermediate-risk AML younger than 60 years, *PHF6*[MT] conveyed worse prognosis[12].

Here, we show three main findings in our comprehensive genomic and proteomic study. Firstly, we show a different pattern of genes correlating with *PHF6*[MT] in male and female cases. When analyzing male and female cases separately, in only male cases, *RUNX1* and *U2AF1* are co-mutated with *PHF6*. In contrast, female cases reveal co-occurrence of *ASXL1* mutations and X chromosome deletions with *PHF6*[MT]. Next, proteomics analysis reveals a direct interaction between PHF6 and RUNX1. Both proteins co-localize in active enhancer regions that define the context of lineage differentiation. Finally, we demonstrate a negative prognostic role of *PHF6*[MT], especially in association with *RUNX1*. The negative effects on survival are additive as *PHF6*[MT] cases with *RUNX1* mutations have worse outcomes when compared to cases carrying single mutation or wild-type.

## Results

### Quantitative aspects of *PHF6*[MT] in MN cases
We started with the analysis of the topography of *PHF6*[MT] in 8443 MN cases (Table 1 and Supplementary Data 1). While frameshifts were found across all the coding regions, missense and stopgains were concentrated in 2 PHD-type zinc finger domains and overlapped with the previously reported MN and T-ALL cases (Fig. 1A)[2,3]. The ePHD2 domain would be essential for *PHF6* function because 88% of non-synonymous or in-frame mutations concentrated in the ePHD2 domain. This is in accordance with a previous study[13]. As the ePHD2 domain is responsible for chromatin binding through recognition of acetylated histones, mutations in ePHD2 domain causing loss of ePHD2 domain would affect chromatin occupancy with *PHF6*. To support our hypothesis, we have also found that >90% of stopgain, frameshift, and splice site mutations are located anteriorly in the ePHD2 domain. The distribution of mutations between MN entities was similar, and there was no difference between male and female cases (Supplementary Fig. 1A, B). According to age, *PHF6*[MT] appeared less common in younger (20–40 years) patients (Fig. 1B). *PHF6*[MT] was significantly more common in male cases with AML (2.3 *vs.* 0.8% of all cases with M/F ratio of 3.5; *p* < 0.001; Fig. 1C). *PHF6* is located on X chromosome, we used the cell fraction instead of the variant allele frequency. Notably, sAML showed higher frequency of *PHF6*[MT] than primary AML (pAML; 2.9% vs. 1.4%; *p* < 0.001; Fig. 1D). For MDS, myelodysplastic/myeloproliferative neoplasm (MDS/MPN), and MPN (myeloproliferative neoplasm), the numbers of cases were too small to determine sex differences (Supplementary Fig. 2A, B).

We then focused on AML in which there was a clear male predilection. When we analyzed male female-skewing for patients with other X chromosomal mutations, we noted three distinct gene groups of mutations: (i) higher frequency of male (*UBA1*, *ZRSR2*, and *STAG2*), (ii) similar frequency in both sexes (*BCORL1*, *BCOR*, *ATRX*, and *PIGA*), and (iii) higher frequency of mutations in female (*KDM6A*; Fig. 2A and Supplementary Table 1). *UBA1*, *ZRSR2*, and *KDM6A* are known as escaping genes from X chromosome inactivation[14] and indeed these genes were significantly overexpressed in females (Fig. 2B), while promoter methylation levels were equal between sexes (delta beta

values < 0.20; Fig. 2C). Consequently, mutant genes escaping from X chromosomal inactivation were highly expressed in females, had similar methylation levels in promoter regions, and mutated at a lower frequency in female cases. In contrast, non-escaping genes in female had similar expression levels, higher methylation levels in promoter regions, and similar frequency of mutations. *PHF6* showed expression and mutational pattern of partially escaping genes. When we evaluated the relationship between the frequency of *PHF6*[MT] and deletion of X chromosome (delX) in female patients, females carrying delX showed a higher frequency of *PHF6*[MT] compared to female cases with diploid X chromosome (6.5% vs. 0.8%, *p* = 0.0019; Fig. 2D). Consequently, when delX were included, a total of 2.3% of female patients had *PHF6*[MT] with 10% of them in a homo/hemizygous configuration (Supplementary Fig. 3). Notably, other non-escaping genes did not reveal higher frequencies of mutations in delX female cases (Supplementary Fig. 4).

### Genomic landscape of *PHF6*[MT]
We next evaluated other co-occurring genomic abnormalities in *PHF6*[MT] MN cases (Supplementary Data 2). Irrespective of the disease type, *ASXL1*, *RUNX1*, and *U2AF1* were most commonly co-mutated with *PHF6* (Fig. 3A). *PHF6*[MT] was significantly more prevalent among cases with *RUNX1*[MT] compared to MN cases without *RUNX1*[MT] (4.4 vs. 1.4%; *p* < 0.001) and significantly co-occurred with *RUNX1*[MT] only in male cases (Supplementary Fig. 5). In addition, *PHF6*[MT] was significantly more frequent in cases with *RUNX1*[MT] in AML and MDS (*p* < 0.001 and *p* = 0.032, respectively). When male and female patients were analyzed separately, the relationship with *RUNX1* and *U2AF1* was found only in male patients (Fig. 3B, Supplementary Fig. 6A–C). In contrast, female cases revealed co-occurrence of *ASXL1* mutations, and *PHF6*[MT] coincided with *RUNX1*[MT] only in females with delX. As expected, combinations of mutations including X chromosomal genes were found more commonly in male patients (Fig. 3C). In sAML, *PHF6*[MT] were enriched in male cases with sAML and *RUNX1*[MT] (2.9% of all sAML, 3.7% of male sAML, 9.2% of male sAML with *RUNX1*[MT]) and vice versa all female cases with sAML, *RUNX1*[MT] and delX also acquired *PHF6*[MT] (Fig. 3D).

A higher frequency of *PHF6*[MT] in advanced MN rather than early diseases (e.g., low-risk MDS) suggested that *PHF6*[MT] is a secondary rather than a founder hit. Clonal architecture analysis using variant allele frequency (VAF) method demonstrated that *RUNX1*[MT] were dominant/co-dominant to *PHF6* hits in the clonal evolution of AML in contrast to T-ALL, in which *PHF6*[MT] appears to be an earlier event (Fig. 3E). Similar relationship was found with regard to *PHF6* hits and other AML-associated gene mutations (e.g., *ASXL1*, *DNMT3A*, or *SRSF2*) suggesting a lineage-restricted tumor suppressor function of *PHF6* in AML following myeloid commitment (Fig. 3F).

### Clinical impact of *PHF6*[MT] on AML cases
To assess the relationship between *PHF6*[MT] and outcome in AML, we first analyzed the clinical impact of *PHF6*[MT] in all patients followed by sub-analysis in male *vs.* female cases. With a median follow-up of 13.0 months (range 0.1–178 months), *PHF6*[MT] AML cases had a shorter overall survival (OS) when compared to wild-type (WT) cases (28% vs. 42% at 3 years, respectively; *p* = $3.8 \times 10^{-4}$; Fig. 4A). Each cohort revealed the similar tendency. Although there were no significant differences in survival in Cleveland Clinic Foundation (CCF) and open data cohort, respectively, the combined CCF and open data cohort showed significantly shorter overall survival in *PHF6*-mutated cases (Supplementary Fig. 7A–D). It is important to mention that when accounting for sex, only male patients with *PHF6*[MT] AML had significantly shorter OS (26% vs. 40% at 3 years, respectively; *p* = $2.9 \times 10^{-3}$; Fig. 4B), whereas no clinical impact was noted in female cases (Fig. 4C). Using our AML cohorts with available leukemic relapse information, event-free survival (EFS) was also shorter in *PHF6*[MT] cases (10% *vs.* 32% at 3 years, respectively; *p* = $3.6 \times 10^{-4}$; Supplementary Fig. 8A). Similar

**Table 1 | Patients' clinical characteristics**

| Characteristics | | | | *PHF6* mutation (%) |
|---|---|---|---|---|
| Number of patients | | | 8443 | 1.74 |
| Age | Median (range) | | 68 (0.4-100) | |
| | ≤60 yr | | 2356 | 1.23 |
| | >60 yr | | 4761 | 1.68 |
| | NA | | 1326 | |
| Sex | Male (%) | | 4603 | 2.37 |
| | Female (%) | | 3840 | 0.94 |
| Disease | AML | pAML | 5784 | 1.24 |
| | | sAML | 1102 | 2.72 |
| | MDS | SLD | 71 | 4.23 |
| | | RS | 227 | 1.32 |
| | | MLD | 147 | 1.36 |
| | | EB | 313 | 2.88 |
| | | del5q | 109 | 0.00 |
| | | Other or NA | 60 | 0.00 |
| | MDS/MPN | CMML | 157 | 3.18 |
| | | aCML | 23 | 4.35 |
| | | MDS/MPN-RS-T | 66 | 1.52 |
| | | Other or NA | 140 | 2.86 |
| | MPN | CMV | 30 | 0.00 |
| | | PV | 57 | 3.51 |
| | | PMF | 28 | 0.00 |
| | | ET | 56 | 1.79 |
| | | Other or NA | 73 | 0.00 |

to OS, only *PHF6*[MT] male cases showed significantly shorter EFS (8.5% vs. 30% at 3 years, respectively; $p = 2.2 \times 10^{-3}$; Supplementary Fig. 8B, C). According to ELN2017, *PHF6*[MT] conveyed a significantly shorter OS only within the adverse risk group, (11% vs. 25% at 3 years, respectively; $p = 2.6 \times 10^{-4}$; Supplementary Fig. 9A). However, in such a setting, the presence of *PHF6*[MT] affected both sexes (*PHF6*[MT] vs. WT at 11% vs. 22% at 3 years, $p = 4.9 \times 10^{-3}$ in males; 9.0% vs. 29% at 3 years, respectively; $p = 0.027$ in females; Supplementary Fig. 9B, C). In contrast, no survival differences were observed between *PHF6*[MT] and WT cases in favorable and intermediate-risk groups (Supplementary Fig. 9D–I). While *RUNX1* mutant AML is categorized to the adverse risk ELN group, *PHF6*[MT] exerted an additional negative effect on survival with *RUNX1*[MT] when compared to cases carrying these mutations alone or WT both in male and female patients ($p = 3.5 \times 10^{-18}$ in all; $p = 6.9 \times 10^{-15}$ in male; $p = 9.0 \times 10^{-4}$ in female; Fig. 5A, C). Munich Leukemia Laboratory (MLL) and open cohort also showed the same result. However, it was not significant in CCF cohort. It would be due to inferior outcome because CCF cohort included higher proportion of sAML (Supplementary Fig. 10A, C). Overall, in univariate and multivariate analyses, *PHF6* was found to be a negative risk factor for survival, regardless of the presence of other genetic mutations including *RUNX1* (hazard ratio [HR] = 1.76, 95% confidence interval [CI] = 1.32–2.33; $p = 9.4 \times 10^{-5}$; Table 2). In accordance with OS, the double-mutated cases revealed worse EFS ($p < 0.0001$ in all; $p < 0.0001$ in male; $p = 0.012$ in female; Supplementary Fig. 11A–C). A detailed analysis of the double-mutated cases showed that most of these cases did not relapse, but did not achieve remission. Thus, conventional therapies would be not effective for these double-mutated cases.

**Proteomics analysis revealed PHF6 interacting partner−RUNX1**

In order to characterize the function of *PHF6*, we performed unbiased interactome by proteomic analysis using immunoprecipitations coupled with LC-MS/MS in the AML cell line THP-1 (derived from a male patient; wild-type configuration of *PHF6* and *RUNX1*). Endogenous PHF6 interactome enriched with chromatin remodelers, SWI/SNF, and NuRD, supporting the role of PHF6 in chromatin organization and DNA repair proteins such as BLM and MSH2 in AML (Fig. 6A and Supplementary Data 3) as well as T-ALL[7]. Interestingly, PHF6 was also associated with several proteins in DNA repair and mRNA splicing, such as MSH2, BLM, SRRT, or DDX5. PHF6 plays a role in chromatin remodeling, replication, and DNA repair not only in T-ALL but also in AML. However, in contrast to T-ALL, hematopoietic lineage-defining pioneer transcription factors, including RUNX1, CBFB, and SPI1 were among the most noticeable and functionally relevant co-immunoprecipitated proteins. We confirmed PHF6 and RUNX1 interaction with each other in normal physiological conditions with reciprocal IP western blot using the THP-1, mouse spleens, and bone marrow cells (Fig. 6B, C and Supplementary Fig. 12A, B). These results support the notion that the interaction between PHF6 and RUNX1 is present in various cell types and contexts, including both normal physiological conditions and disease states. The inclusion of AML cells (THP-1) in our study adds relevance to the context of myeloid malignancies, as it demonstrates the persistence of the PHF6-RUNX1 interaction in a cell line representative of the disease. Consistent with our hypothesis and co-immunoprecipitation experiments, ChIPseq showed the co-localization of PHF6 and RUNX1 in multiple regions, including active enhancers (Fig. 6D, E and Supplementary Data 4–6). We further examined changes in gene expression patterns associated with *PHF6*[MT]. In *PHF6*[MT] *vs.* WT AML, several genes specific to lymphoid lineages were highly expressed (Fig. 6F and Supplementary Data 7), including *DNTT* expressed only in normal and malignant lymphoid cells. In contrast, several myeloid-specific genes, such as *CD33* or *CSF2RB* were downregulated. Consistent with this result, gene set enrichment analysis (GSEA) demonstrated upregulation of lymphocyte differentiation genes while myeloid markers were downregulated (Fig. 6G, Supplementary Table 2, and Supplementary Data 8). However, while upregulating lymphoid gene sets (including B-cell differentiation), *RUNX1*[MT] showed a decreased expression of myeloid markers as opposed to WT AML cases (Supplementary Fig. 13A, B, Supplementary Data 9, 10, and Supplementary Table 3). This result is consistent with the previously described role of *RUNX1*[MT] in mediating lymphoid differentiation[15]. To illustrate this point, we selected several genes overexpressed in *PHF6*[MT] AML (high clonal *PHF6* burden) and performed immunohistochemistry (IHC) staining with specific antibodies. IHC and immunophenotype analysis (Supplementary Table 4) showed positivity for LY9, or GCSAM, and/or TdT in 2/6 *PHF6*[MT] cases (Fig. 6H–I). In 4/6 of these cases, *PHF6*[MT] coincided with *RUNX1*[MT]. Altogether, these findings suggest that PHF6 co-localizes with RUNX1 in the nucleus and would cooperate in the differentiation of blood cells along myeloid or lymphoid lineages.

## Discussion

*PHF6*[MT] are important mutations in both lymphoid and myeloid leukemias. Our study has found both similarities and differences in terms of the molecular background, consequences, and clinical impact of *PHF6*[MT] in these disease groups.

Firstly, we showed that there is a male predilection for the acquisition *PHF6*[MT] in AML consistent with a partially incomplete X chromosomal inactivation status, although a significant minority of females may also be affected by these mutations. In such cases, *PHF6*[MT] was also significantly associated with delX indicating that its biallelic inactivation assures functional consequences due to incomplete X chromosomal inactivation. A similar role of delX has also been seen in female patients with pediatric T-ALL and AML[16,17]. As in *PHF6*[MT] male cases, the selection pressure would likely cause mutations in other genes mapping on X chromosome. While mutation analysis suggests that *PHF6* showed an incomplete inactivation pattern, expression, and DNA methylation studies rather indicated complete inactivation. One could stipulate that escaping from X chromosome inactivation is

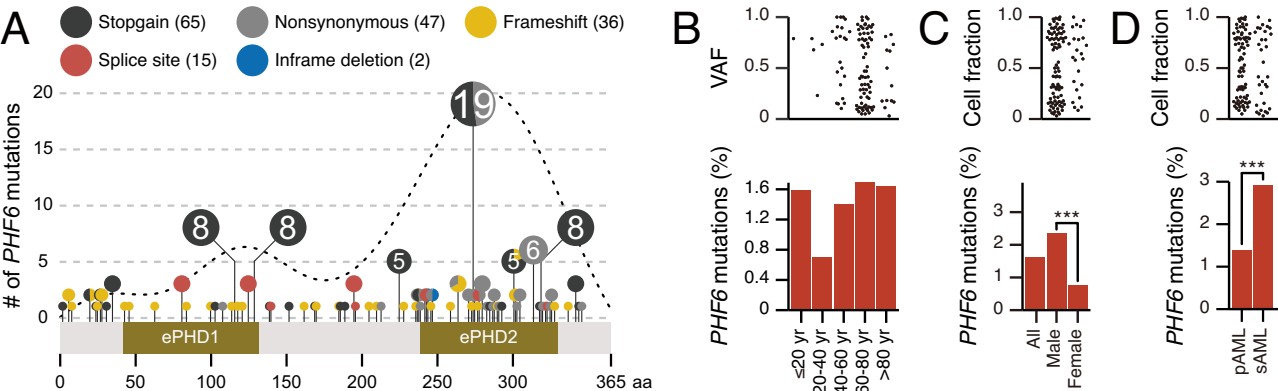

**Fig. 1 | The frequencies and positions of *PHF6* mutations. A** Lollipop plot illustrating *PHF6* mutational spectrum in this study group. Mutational subtypes are shown by colors as indicated. The numbers in circles indicate the number of cases. **B** Comparisons of frequencies of *PHF6* mutations based on patients' age groups (≤20 years, $n = 63$; 20–40 years, $n = 577$; 40–60 years, $n = 1716$; 60–80 years, $n = 4090$, >80 years, $n = 671$). Each dot in the upper panel represents the variant allelic frequencies with *PHF6* mutations. **C** Comparisons of frequencies of *PHF6* mutations in each sex of AML cases ($n = 3715$ and 3172, respectively). Each dot in the upper panel represents the cell fractions with *PHF6* mutations. **D** Comparisons of frequencies of *PHF6* mutations in primary and secondary AML cases ($n = 5785$ and 1102, respectively). Each dot in the upper panel represents the cell fractions with *PHF6* mutations. **C**, **D** $p$ values were calculated by two-sided Fisher's exact test. *$p < 0.05$, **$p < 0.01$, ***$p < 0.001$. Raw $p$ values are as follows: $p = 1.6 \times 10^{-9}$ (male vs. female AML) and $6.4 \times 10^{-4}$ (primary vs. secondary AML).

situational i.e., lineage-specific differentiation. By analogy, although *STAG2* and *ATRX* are non-escaping genes in hematopoietic stem cells, they show biallelic expression in female brain cells and fibroblasts, respectively[14,18]. Thus, biallelic *PHF6* might be specifically expressed in immature myeloid cells. Besides, survival analysis suggests that in female patients with *PHF6*[MT] AML only monoallelic lesions would be functional upon AML development.

Our study also showed that PHF6 mutually interacts with RUNX1, providing a mechanistic rationale for their frequent mutational co-occurrence. In this study, we revealed the interaction between wild-type PHF6 and RUNX1. In case of the presence of *PHF6*[MT], such inter-action would be lost. Because in our investigation, we observed that the genetic alterations in *PHF6* predominantly involved stopgains, frameshifts, and splice site mutations resulting in nonsense-mediated decay or producing abnormal proteins due to truncation of abnormal splicing (Supplementary Fig. 14). This indicates that PHF6 interaction with RUNX1 would be lost in males with *PHF6*[MT] and in females with *PHF6*[MT] and delX. Western blotting for AML samples with frameshift or stopgain mutations in *PHF6* showed low expression reiterating the described conclusion (Supplementary Fig. 15). *RUNX1*-mutated samples with Runt domain nonsynonymous mutations accounted for 98% of nonsynonymous *RUNX1* mutations in our cohort. IP Western analysis demonstrated that Runt domain mutation resulted in a less abundant co-immunoprecipitation of PHF6 than samples with wild-type *RUNX1* (Supplementary Fig. 16). This result suggested that RUNX1 interacts with PHF6 via Runt domain also fitting with the conclusion that *PHF6* mutant samples have no or weaker interaction between PHF6 and RUNX1. The fact that most of the *PHF6* mutations are present in cases with *RUNX1* truncated after Runt domain also support the notion that the interaction between RUNX1 and PHF6 involves Runt domain (supplementary Fig. 17). Indeed, our results illustrated co-localization of PHF6 with RUNX1 in active enhancer regions likely via direct PHF6 and RUNX1 interaction. As a consequence, this coopera-tion would induce differentiation to myeloid lineage and suppress differentiation to lymphoid lineage. This was in line with a previous report demonstrating that PHF6 co-localizes with RUNX1 in promoter regions[19]. The chromatin-binding capacity of PHF6 is essential for differentiation function in AML, as shown in B-ALL[9]. Indeed, stopgain and nonsynonymous mutations were concentrated in zinc finger domains, which are important for the function of chromatin-binding capacity in PHF6. In myeloid cells, we have also demonstrated using immunoprecipitation that PHF6 binds to many chromatin-binding

proteins such as SWI/SNF and NuRD complexes. A particular remark to understand this finding is that *PHF6* was more frequent in sAML along with *RUNX1*. While present together in many cases, the order in which *PHF6* and *RUNX1* mutations occur is important. Once acquired *RUNX1*[MT], tumor cells would suffer perturbation of the chromatin structures[20], and subsequently undergo differentiation block. In this situation, *PHF6* may be partially able to compensate *RUNX1* function, ultimately leading to a down progression. This process could be then relieved by a subsequent acquisition of *PHF6*[MT] and loss of chromatin-binding capacity, completely blocking differentiation and responsible for the development of AML with adverse risk features. Such a mechanism may cause myeloid skewing in tumor cells, as shown by analysis of other myeloid genes. Conversely, if acquiring *PHF6*[MT] first, T-ALL or MPAL might develop as indicated by the rather subclonal presence of *RUNX1*[MT] *vs. PHF6*[MT] in several T-ALL or MPAL cohorts[21,22].

Finally, we were able to demonstrate a negative prognostic role of *PHF6*[MT] especially when co-mutated with *RUNX1*. Comparing double-mutant (*PHF6* and *RUNX1*) and single-mutant (*PHF6* or *RUNX1*) cases, there was no difference of mutated domains or mutational types between double-mutated and single-mutated cases (Supplementary Fig. 18A–D). It is similar to *RUNX1* mutations that concentrated in the Runt domain in both double-mutated and single-mutated cases. Since *PHF6* has a complementary function to *RUNX1*, it can be postulated that co-mutation of these two genes may show an additive detrimental effect on prognosis in addition to losing the tumor-suppressive role of *PHF6*. Thus, as shown by our analysis *PHF6*[MT] cases with *RUNX1*[MT] should be classified as a subset of AML with very poor survival out-comes. However, long-term survival may be overestimated in our cohort, because of the short median follow-up period. In *PHF6*-muta-ted AML cells, as IHC studies revealed, surface proteins such as LY9 or GCSAM induced by *DNTT* expression can be targetable. However, since higher mRNA expression does not necessarily correlate with higher protein expression in AML cells[23], further studies are warranted to fully elucidate the full proteomic spectrum of *PHF6*-mutated AML. This strategy may reveal specific therapeutic vulnerabilities potentially applicable also in other contexts, such as T-ALL and B-ALL.

By examining the interaction between wild-type PHF6 and RUNX1, we aimed to establish a baseline understanding of their functional relationship and the mechanistic implications of their interaction in cellular processes. We believe that comprehending the wild-type interaction is foundational for future investigations to evaluate the consequences of mutations in both PHF6 and RUNX1 on their

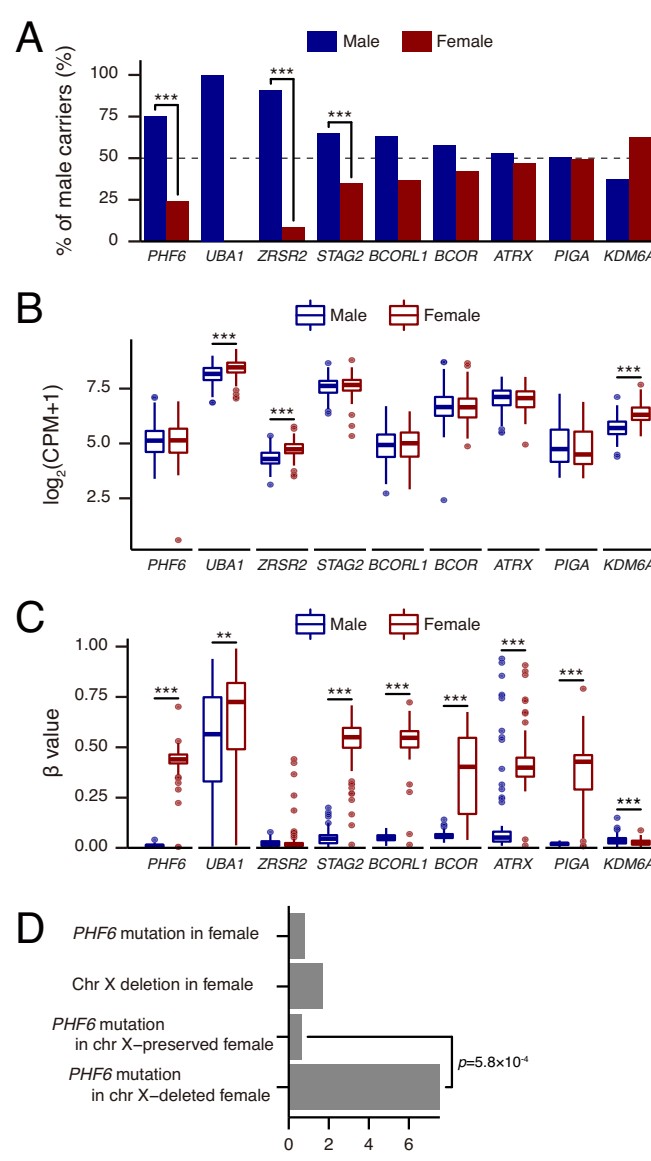

**Fig. 2 | The frequencies of mutations of genes on X chromosome and association with DNA methylation, expression, and X chromosome. A** Comparisons of frequencies of mutations in genes on X chromosome in AML cases ($n = 6887$). The genes are ordered by mutation rate in male cases. The frequencies of *UBA1* and *PIGA* mutations were based on other our cohorts. Raw *p* values are as follows: $p = 2.2 \times 10^{-6}$ (*PHF6*), $1.2 \times 10^{-5}$ (*ZRSR2*), and $4.7 \times 10^{-4}$ (*STAG2*). **B** Comparisons of mRNA expression levels in genes on X chromosome between male ($n = 139$) and female ($n = 113$) cases in the Beat AML cohort. The *p* value was calculated using the two-tailed Student's *t* test. The mean, 25th, and 75th percentiles are represented in the box plots by the midline and box edges, respectively. The whiskers extend to 1.5 times the interquartile range. Dots represent each outlier of expression. Raw FDR are as follows: FDR = $9.3 \times 10^{-7}$ (*UBA1*), $3.2 \times 10^{-16}$ (*ZRSR2*), and $8.5 \times 10^{-25}$ (*KDM6A*). **C** Comparisons of DNA methylation levels in the promoter regions of genes mapping on X chromosome between male ($n = 103$) and female ($n = 99$) cases in the Beat AML cohort. The mean, 25th, and 75th percentiles are represented in the box plots by the midline and box edges, respectively. The whiskers extend to 1.5 times the interquartile range. Dots represent each outlier of DNA methylation. Raw FDR are as follows: $p = 2.9 \times 10^{-72}$ (*PHF6*), $1.0 \times 10^{-3}$ (*UBA1*), $7.2 \times 10^{-63}$ (*STAG2*), $2.0 \times 10^{-73}$ (*BCORL1*), $1.3 \times 10^{-27}$ (*BCOR*), $2.9 \times 10^{-25}$ (*ATRX*), $3.4 \times 10^{-44}$ (*PIGA*), and $2.1 \times 10^{-5}$ (*KDM6A*). **D** Comparisons of frequencies of *PHF6* mutations and X chromosome deletions in female AML cases (*PHF6* mutation and X chromosome deletion, $n = 4$; *PHF6* wild-type and X chromosome deletion, $n = 49$; *PHF6* mutation and normal X chromosome, $n = 20$; *PHF6* wild-type and normal X chromosome, $n = 3091$). **A**, **D** *p* values were calculated by two-sided Fisher's exact test and not adjusted for multiple comparison. *$p < 0.05$, **$p < 0.01$, ***$p < 0.001$. **B**, **C**, *p* values were calculated by using two-tailed Student's *t* test and adjusted by Benjamini–Hochberg correction. *FDR < 0.05, **FDR < 0.01, ***FDR < 0.001.

prognosis in AML cases, and co-mutation with *RUNX1* leads devastating outcome. Proteins aberrantly expressed on the cell surface are targetable for molecular-specific therapeutic strategies. In addition to *PHF6*, mutations of tumor suppressor genes in X chromosome represent higher risk features in male patients, calling for a reconsideration of sex bias into risk classifications.

## Methods

### Patients

In total, 8443 patients diagnosed with AML, MDS, MDS/MPN, and MPN were included in this study based on sample and clinical information availability (Table 1 and Supplementary Table 1). Samples were mainly collected from two cohorts, CCF and MLL. Compensation was not provided to patients. Sex of participants was determined based on self-report and chromosomal karyotyping. Our cohort consisted of 1465 CCF, 5109 MLL, and 1869 open data cases (Supplementary Table 5). Open data cases were whole-exome sequencing cases ($n = 618$) from Beat AML[24] and targeted capture sequencing cases ($n = 1251$) from German-Austrian AML study[25]. While we used all the available information regarding the clinical, karyotype, and DNA sequencing data from CCF, MLL, the Beat AML, and the German-Austrian AML study, the frequencies of *UBA1* and *PIGA* mutations in Fig. 2A were based on previously published cohorts[37,38]. Also we used the Beat AML cohort for RNA expression and DNA methylation analyses[24,26]. This research was conducted under the Institutional Review Boards of CCF (IRB #5024) and of MLL (Ethic Komission IRB #05117). Blood and bone marrow samples were collected after written informed consent. All protocols conformed to the tenets of the Declaration of Helsinki. Samples were obtained from peripheral blood and/or bone marrow aspirate. Genomic DNA was isolated with the Nuclei Lysis Solution (Promega, A7941) according to the manufacturer's instructions.

### Next-generation sequencing

For the data collected at CCF, whole-exome sequencing or targeted sequencing was performed on paired tumor and germline DNA (purified CD3+ lymphocytes). Whole-exome capture was accomplished according to the SureSelect Human All Exon 50 Mb or V4 Kit (Agilent Technologies, CA, USA). Targeted sequencing was performed as

interaction dynamics, as well as the potential disruption of chromatin occupancy. However, our study has several limitations. First, we were not able to show which domain in PHF6 is important for the interaction with RUNX1 or chromatin. As shown through our mutational analysis, we speculated that ePHD2 domain is essential for PHF6 function. Consequently, the precise clarification of the interaction principles between wild-type and mutant RUNX1/PHF6 may require a separate analysis involving individual mutant types/locations of RUNX1, DNA binding, and analysis of the corepressor interaction and assessment of the wild-type allele effects. Second, it remains to be precisely elucidated which genomic regions PHF6 co-occupy with RUNX1. However, our ChIPseq result showed PHF6 and RUNX1 bind at multiple regions, including active enhancers. While possible for some sites, it is unlikely that PHF6 and RUNX1 proteins bind independently to all the regions occupied by these proteins. Finally, we were not able to show why *PHF6* and *RUNX1* mutations have additive effect. This would be due to the differentiation function of PHF6 and RUNX1, and the tumor-suppressive function of PHF6. To unveil this additive effect, we need to elucidate the functional consequences of these mutations on protein-protein interactions and downstream cellular processes.

In conclusion, here we demonstrate that PHF6 cooperates with RUNX1 and regulates cell lineage differentiation. *PHF6* also affects

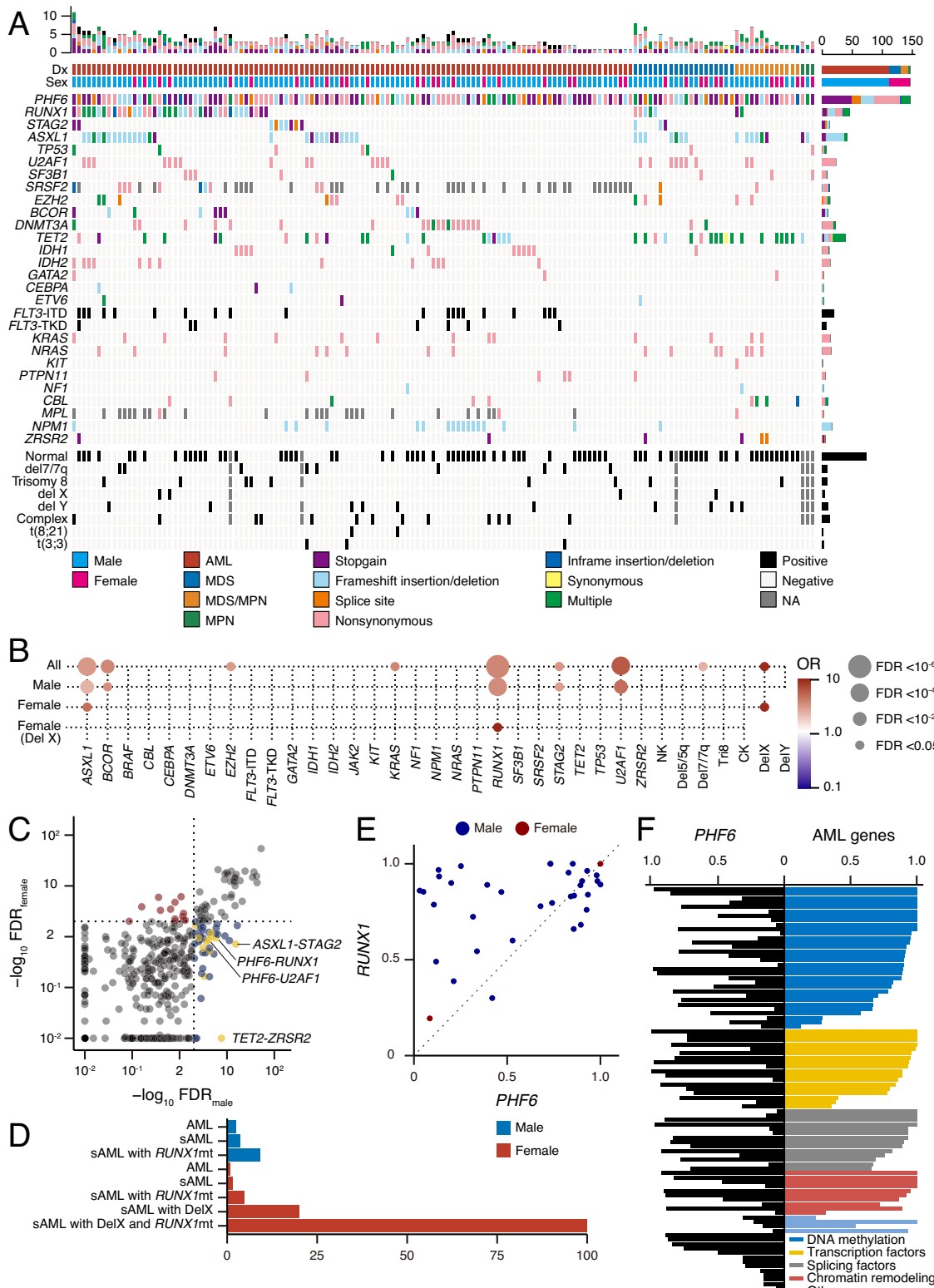

**Fig. 3 | The correlation of *PHF6* mutations with other genetic mutations.**
**A** Genetic landscape of *PHF6*-mutated MN. Mutational subtypes are shown by colors as indicated. **B** Bubble plot showcasing statistically significant (FDR < 0.05) positive (red) and negative (blue) correlations across gene group in all (*n* = 6887), male (*n* = 3715), female (*n* = 3172), and female with X chromosome deletion (*n* = 53). **C** The significance in combinations of mutated genes between male and female

AML (*n* = 3715 and 3172, respectively). **D** The frequencies of *PHF6* mutations based on each AML subtype. **E** The cell fraction with *PHF6* versus *RUNX1* mutations in AML. **F** Each horizontal bar represents the cell fractions with mutations of *PHF6* and other AML-associated genes in each case. **B** *p* values were calculated by two-sided Fisher's exact test and adjusted for multiple comparisons with Benjamini−Hochberg adjustment.

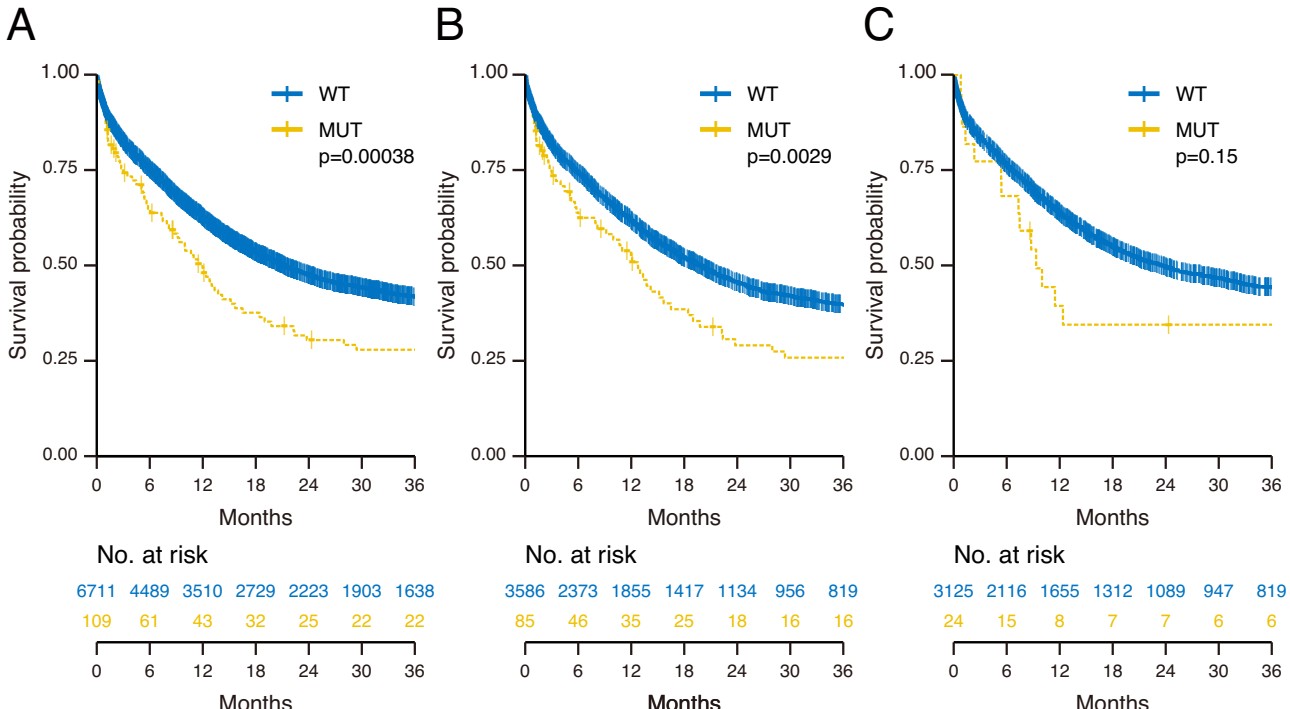

**Fig. 4 | Clinical impact of *PHF6* mutations in AML. A** Kaplan–Meier survival curves of overall survival for all AML cases with and without *PHF6* mutations (*n* = 109 and 6711, respectively). **B** Kaplan–Meier survival curves of overall survival for male AML cases with and without *PHF6* mutations (*n* = 85 and 3586, respectively). **C** Kaplan–Meier survival curves of overall survival for female AML cases with and without *PHF6* mutations (*n* = 24 and 3125, respectively). For survival analysis, survival was estimated using the Kaplan–Meier method, and the log-rank test was used to assess differences between groups.

previously described using a custom panel from TruSeq or Nextera platforms (Illumina, San Diego, CA, USA). Sequencing libraries were generated according to an Illumina paired-end library protocol and sequenced using a HiSeq 2000/2500/X (Illumina, San Diego, CA, USA). Paired-end sequenced reads were aligned to the human genome (hg19) with the Burrows-Wheeler Aligner (http://bio-bwa.sourceforge.net/) to GRCh37 reference, and post-alignment processing included sorting, marking of duplicates, indexing, base recalibration, according to Genome Analysis Tool Kit v.3 best practices. Variants were annotated using Annovar and filtered by in-house bioanalytic pipeline[27]. Libraries in patients from the MLL cohort were generated using the TruSeq PCR-Free prep kit according to the manufacturer's recommendations (Illumina, San Diego, CA) and sequenced on NovaSeq6000/HiSeqX Illumina instruments by following a 2 × 150-bp paired-end–reads standard protocol. Data were analyzed on Illumina's BaseSpace Sequence Hub and in-house pipelines. Reads were aligned against human genome build 19 (hg19) with the tool Isaac3[28]. Variant calling was performed using Strelka2[29] and variants were annotated with Ensembl VEP30[30]. For the purpose of this study, only exonic (nonsynonymous single-nucleotide variants and small insertions/deletions) and splicing variants were considered.

### Variants panel and filtering
Detected variants were filtered using the following criteria: (i) only variants with a minimum depth of 10 reads and 4 reads supporting the alternate allele were considered; (ii) synonymous, polymorphisms (global population frequency >1%), and potential germline variants, or mapping errors (visual inspection with the Integral Genomics Viewer[31]) were removed; (iii) nonsynonymous variants (missense, nonsense, frameshift, and indels) were included, and further filtered according to sequenced controls such as gnomAD database (https://grnomad.broadinstitute.org/) (≤0.1%), dbSNP13850[32], 1000Genomes[33], and mutational databases including COSMIC[34] and ClinVar[35] for pathogenicity confirmation. Finally, filtered variants were classified as pathogenic/likely pathogenic, variants of unknown significance, benign/likely benign variants based on ACMG/AMP criteria[36]. Only pathogenic and likely pathogenic variants were used for the purpose of this study to increase stringency in terms of clinical consequences. The frequencies of *UBA1* and *PIGA* mutations were based on other cohorts, respectively[37,38].

### Cell fractionation and nuclear protein extraction
Approximately 100 million THP-1 cells, 25 million primary AML cells, and 400 million mouse spleen and bone marrow cells (from 4 mouse) were used in PHF6 immunoprecipitations. Cells were transferred to 15-mL conical tubes and washed twice with 10 mL ice-cold 1× PBS that contained protease inhibitors (Sigma-Aldrich, A8340). Cells were resuspended in 2 mL of 1× hypotonic buffer containing 10 mM N-2-hydroxyethylpiperazine-N′–2-ethanesulfonic acid, 1.5 mM MgCl2, 10 mM KCl, 0.5 mM dithiothreitol, 10 mM PMSF, and protease inhibitors (Sigma-Aldrich, A8340). A total of 40 µL of 10% NP-40 was added to cell suspensions to break the cell membrane. After 5-minute incubation on ice, cell suspensions were centrifuged at 344 g for 10 minutes. The supernatant was transferred to clean 15 mL centrifuge tube and labeled as the cytoplasmic fraction. Nuclear pellets were washed twice with ice-cold 1× PBS, and resuspended in 100 µL of 50 mM Tris-HCl, pH 8.0, 1 mM MgCl₂, 10 mM PMSF, protease inhibitor cocktail (Sigma-Aldrich, A8340) and Benzonase (Sigma-Aldrich, D5915, 250 units). The nuclear suspensions were incubated on ice for 90 minutes with a vigorous vortex every 10 minutes. At the end of incubation, 400 µL protein extraction buffer containing 2% NP-40, 500 mM NaCl, 5 mM dithiothreitol, 10 mM PMSF, and 5 µL of protease inhibitor cocktail (Sigma-Aldrich, A8340) in 1× PBS buffer (pH 7.4) were added. After 30-minute incubation on ice with vortexing every 5 minutes, the mixture was centrifuged at 12396 g for 15 minutes. The same extraction process was repeated two more time with 500 µL extraction buffer containing 1% NP-40 5 mM dithiothreitol, 10 mM PMSF, and 5 µL of protease inhibitor cocktail (Sigma-Aldrich, A8340) in 1× PBS buffer (pH

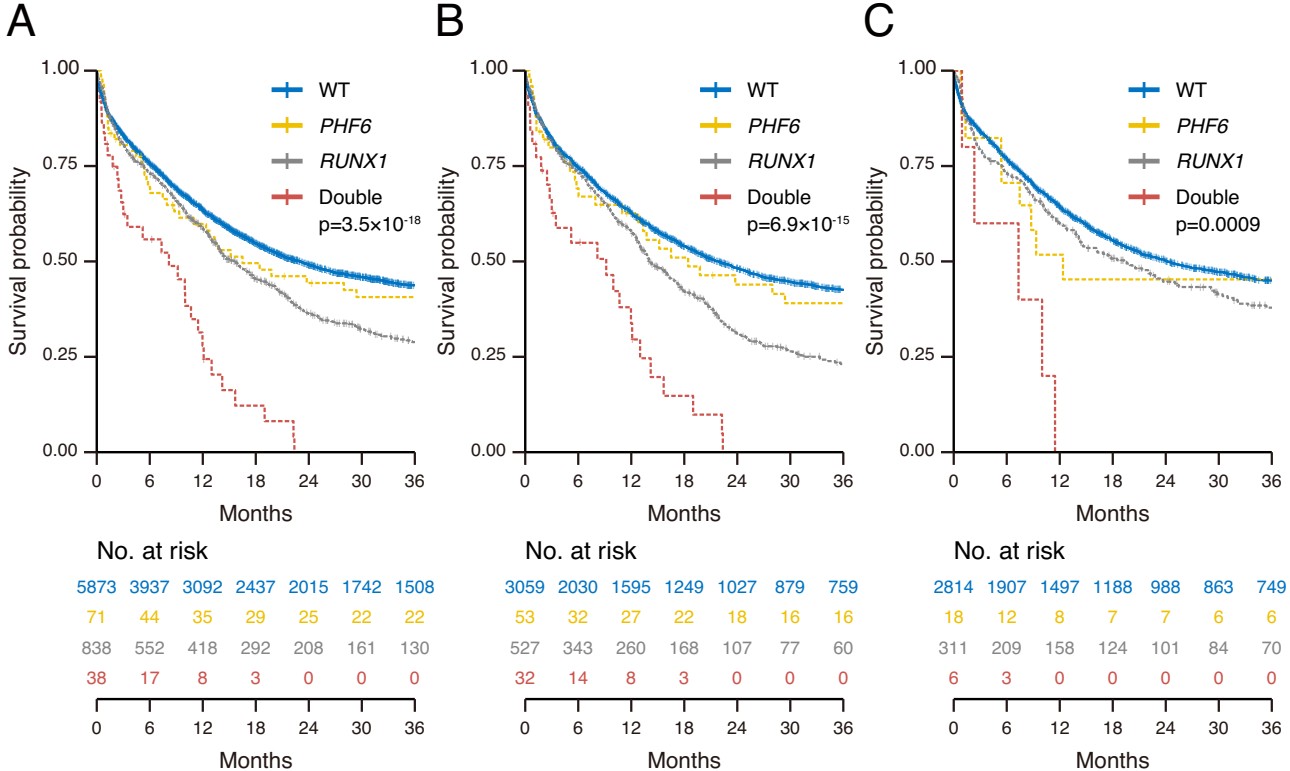

**Fig. 5 | Clinical impact of *PHF6* and *RUNX1* mutations in AML. A** Kaplan–Meier survival curves of overall survival for all AML cases with double mutations (*PHF6* and *RUNX1*; *n* = 38), single mutations (*PHF6*, *n* = 71; *RUNX1*, *n* = 838), and negative cases (*n* = 5873). **B** Kaplan–Meier survival curves of overall survival for male AML cases with double mutations (*PHF6* and *RUNX1*; *n* = 32), single mutations (*PHF6*, *n* = 53; *RUNX1*, *n* = 527), and negative cases (*n* = 3059). **C** Kaplan–Meier survival curves of overall survival for female AML cases with double mutations (*PHF6* and *RUNX1*; *n* = 6), single mutations (*PHF6*, *n* = 18; *RUNX1*, *n* = 311), and negative cases (*n* = 2814). For survival analysis, survival was estimated using the Kaplan–Meier method, and the log-rank test was used to assess differences between groups.

### Table 2 | Univariate and multivariate analyses

| | | Univariate | | | Multivariate | | |
|---|---|---|---|---|---|---|---|
| | | HR | 95% CI | *p* value | HR | 95% CI | *p* value |
| *PHF6* | Mut vs. WT | 1.52 | 1.20–1.91 | $4.16 \times 10^{-4}$ | 1.76 | 1.32–2.33 | $9.42 \times 10^{-5}$ |
| *RUNX1* | Mut vs. WT | 1.42 | 1.30–1.55 | $2.58 \times 10^{-14}$ | 1.11 | 1.00–1.23 | $4.93 \times 10^{-2}$ |
| *U2AF1* | Mut vs. WT | 1.80 | 1.51–2.15 | $9.48 \times 10^{-11}$ | 1.42 | 1.16–1735 | $8.00 \times 10^{-4}$ |
| *ASXL1* | Mut vs. WT | 1.58 | 1.43–1.74 | $8.95 \times 10^{-19}$ | 1.16 | 1.04–1.30 | $8.66 \times 10^{-3}$ |
| Sex | Male vs female | 1.14 | 1.07–1.22 | $5.96 \times 10^{-5}$ | 1.02 | 0.94–1.10 | 0.686 |
| Age | ≤60 yr vs. >60 yr | 2.59 | 2.39–2.82 | $1.89 \times 10^{-111}$ | 2.51 | 2.30–2.73 | $2.95 \times 10^{-100}$ |

7.4). The supernatant containing nuclear proteins was combined and transferred to clean tubes, and protein concentration was determined by BCA assay.

### Immunoprecipitation
Agarose beads conjugated PHF6 (SCBT, sc-365237AC), RUNX1 (SCBT, sc-101146) along with homemade agarose conjugated mouse IgG were used in immunoprecipitation. Nuclear protein extracts were transferred to tubes with antibody-bound protein A/G beads and rocked gently at 4 °C overnight. Nonspecific bound proteins were removed with five washes of 1× PBS containing 1% NP-40. Immunoprecipitation products were extracted from the protein A/G beads using Laemmli sample buffer.

### Protein identification by LC-MS/MS
Anti-PHF6 and isotype antibody immunoprecipitation products were subjected to SDS-polyacrylamide gel electrophoresis and stained with colloidal Coomassie Blue (Gel Code Blue, Pierce Chemical). 8 Gel slices were excised from the top to the bottom of the lane for each sample; proteins were reduced with dithiothreitol (Sigma-Aldrich, D0632, 10 mM), alkylated with iodoacetamide (Sigma-Aldrich, I1149, 55 mM), and digested in situ with trypsin. Peptides were extracted from gel pieces three times using 60% acetonitrile and 0.1% formic acid/water. The dried tryptic peptide mixture was redissolved in 20 μL of 1% formic acid for mass spectrometric analysis. Tryptic peptide mixtures were analyzed by online LC-coupled tandem mass spectrometry (LC-MS/MS) on an Orbitrap mass spectrometer (Thermo Fisher). Initial protein identifications from MS/MS data used Proteome Discovery and Uniprot human protein database containing 85,299 protein sequences. The Swiss Protein database search parameters included 2 missed tryptic cleavage sites allowed, precursor ion mass tolerance of 10 ppm, fragment ion mass tolerance of 0.6 Da, protein modifications for Met oxidation, and Cys carbamidomethylation. Decoy database search was included for false positive estimation. A minimum Xcorr value of 2 and

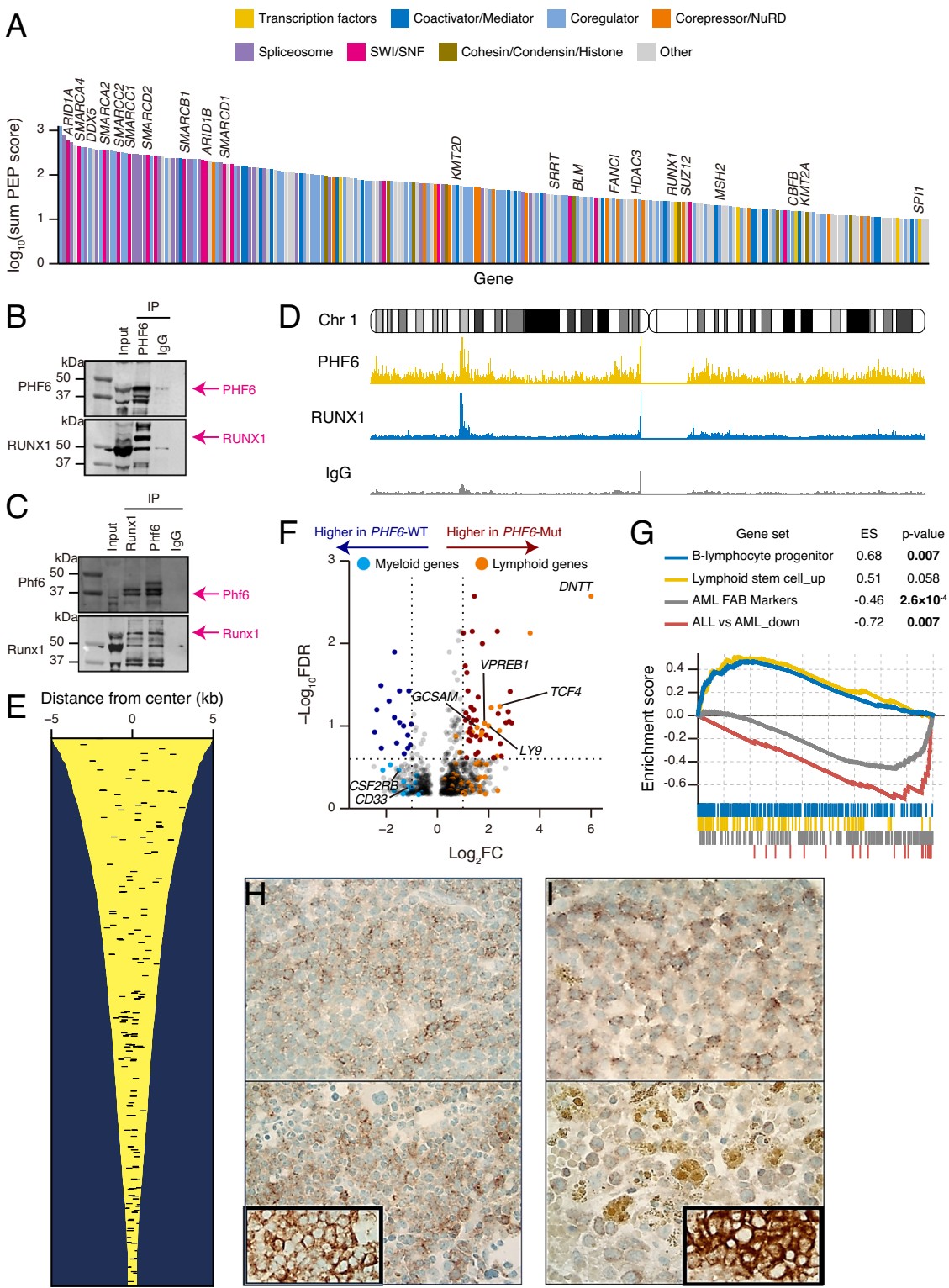

peptide rank 1 were used for automatically accepting peptide MS/MS spectra. Proteins with only one unique peptide sequence (single hits) were also eliminated.

**1D SDS-polyacrylamide gel electrophoresis and western blot analysis.** Immunoprecipitation products, IgG control immunoprecipitation products, and inputs for IP were subjected to 1D SDS-polyacrylamide gel electrophoresis on precast 4% to 12% NuPAGE gels (Invitrogen, NP0335BOX). After electrophoresis per the manufacturer's instructions (Invitrogen), proteins were transferred to polyvinylidene difluoride membranes (Millipore) at 25 constant voltage for 25 min using Trans-Blot® Turbo™ Transfer System (Biorad,1704150). Primary antibody used, Anti-PHF6 Antibody (H-4, SCBT, sc-365237), Anti-RUNX1 Antibody (DW71, SCBT, sc-101146). Loading controls, Anti-β-Actin Antibody (C4) Alexa Fluor® 647 (SCBT, sc-47778 AF647), Anti-GAPDH Antibody (0411) Alexa Fluor® 488 (SCBT, sc-47724 AF488), Anti-Lamin B1 Antibody (B-10) Alexa Fluor® 488 (SCBT, sc-374015 AF488). Secondary antibodies, StarBright Blue 700 Goat Anti-Rabbit IgG (Biorad, 12004161) and StarBright Blue 700 Goat Anti-Mouse IgG (Biorad, 12004158) were used at 1:3000 dilution.

**Fig. 6 | Co-immunoprecipitated proteins with PHF6 and expression profiling in** ***PHF6*-mutated AML. A** Waterfall plot represents the significance of each protein in co-immunoprecipitated proteins with PHF6. Transcription factors, coactivator/mediator, coregulator, corepressor/NuRD, Spliceosome, SWI/SNF, cohesion/condensing/histone, and other proteins are shown by colors as indicated. **B** Western blots of endogenous PHF6 IP of THP-1 nuclear protein extracts. 5% input, PHF6 IP, and IgG control IP product were run side by side. The same membrane was probed with an anti-rabbit monoclonal antibody to PHF6 and an anti-mouse monoclonal antibody to RUNX1. PHF6 and RUNX1 bands are indicated by arrows in pink color. **C** Reciprocal IPs of endogenous Phf6 and Runx1 with mouse spleen and bone marrow protein extracts. 5% input, Runx1 IP, Phf6 IP, and IgG control IP product were run side by side. The same membrane was probed with rabbit monoclonal antibody to Phf6 and mouse monoclonal antibody to Runx1. Phf6 and Runx1 bands are indicated by pink arrows. **D** Normalized distribution of PHF6, RUNX1 ChIPseq,

and IgG control intensities in chromosome 1 in THP-1. **E** PHF6 and RUNX1 co-localization in active enhancer regions. Each yellow line showed H3K27ac enrichment around the peak. Each black bar shows co-localized regions of PHF6 and RUNX1. **F** Volcano plot comparing significant expression difference between *PHF6*-mutated ($n = 7$) and wild-type ($n = 132$) male AML. Lymphoid and myeloid genes with differentially expression are colored orange and blue, respectively. The Bayesian method by the linear models for microarray expression data (limma) package version 3.50.0 in R software. **G** GSEA plot showing changes in lymphoid and myeloid signature genes between *PHF6*-mutated ($n = 7$) and wild-type ($n = 132$) AML. To perform the enrichment of the difference between $PHF6^{MT}$ and WT AML, we used the GSEA software (v.4.3.2). **H** Positive cytoplasmic immunohistochemistry staining of LY9 in blasts in 2 *PHF6*-mutated AML cases. Inlet shows positive control of LY9 (×500). **I** Positive cytoplasmic immunohistochemistry staining of GCSAM in blasts in 2 *PHF6*-mutated AML cases. Inlet shows positive control of GCSAM (×500).

IRDye® 800CW Goat anti-Mouse IgG Secondary Antibody (Li-cor, 926-32210) and IRDye® 800CW Goat anti-Mouse IgG Secondary Antibody (Li-cor, 926-32211) were used at 1:5000 dilution. The Western blot was done twice in THP-1 cells and once in mouse spleen/bone marrow cells, respectively.

### Covalent bound antibody to protein G beads
Mouse control IgG (SCBT, sc-2025) were covalently coupled to Sepharose-protein A/G (SCBT, sc-2003) beads using dimethylpimelimidate (Sigma-Aldrich, D8388). Briefly, 200 μL of Sepharose-protein A/G was washed with 1× PBS twice, incubated with 200 μL of antibody (20 μg) solution (1× PBS) for 1 hour at room temperature. Antibody-bound protein A/G beads were then incubated with 1% chicken egg ovalbumin in PBS for another hour to block nonspecific binding sites. After three washes with 1× PBS, 25 mg of dimethylpimelimidate in 1 mL of 200 mM triethanyl amine was added, and coupling reaction proceeded at room temperature for 30 minutes. The reaction was repeated 2 more times with fresh addition of dimethylpimelimidate and quenched with 50 mM ethanolamine. The reacted protein G beads were washed extensively with 1× PBS before immunoprecipitation.

### RNA expression analysis
For RNA expression analysis, we used the Beat AML cohort[24]. The Bayesian method by the linear models for microarray expression data (limma) package version 3.50.0 in R software[39] was used for the normalization of genes and identification of differentially expressed genes between $PHF6^{MT}$ or $RUNX1^{MT}$ and WT AML cases. The genes with $log_2$ fold change value >1 or <−1, and $−log_{10}$ (false discovery rate)>0.6 were considered as significantly differentially expressed genes.

### ChIPseq
The Zymo-Spin ChIP kit (Zymo Research Corp., Irvine, CA, D5209) and anti-PHF6 antibody (SCBT, sc-365237AC) were used for chromatin immunoprecipitation sequencing (ChIPseq) according to the instructions provided by the manufacturer. For ChIPseq, THP-1 cell lines and a total of 5 million THP-1 cells were resuspended in 1 ml of 1× PBS. The high-throughput sequencing was performed by the Cleveland Clinic Sequencing Core on an Illumina 2500 sequencer using 50 bp single-end sequencing. The Bowtie2 alignment tool was used to align ChIPseq reads to the human genome build hg19. We used the HOMER software for finding peaks in aligned data[40]. Deeptools was used for generating bigwigs to visualize with the Integrative Genomics Viewer (IGV)[31,41]. We downloaded H3K27ac ChIPseq in THP-1 from GEO (GSM5908232) and RUNX1 ChIPseq fastq in THP-1[42]. These downloaded data were also aligned and analyzed as described above.

### Immunohistochemistry
IHC was performed using an automated immunostainer (Discovery Ultra, Ventana Medical Systems, Tucson, AZ). Tissue sections obtained from formalin fixed and paraffin embedded tissue, were subjected to

heat-induced epitope retrieval (Ventana's Cell Conditioning, pH 9.0) for 64 minutes and stained with antibodies against PHF6 (Millipore Sigma, HPA001023, 1:25), LY9 (CD229, Abcam, EPR22611-91 clone, 1:50), and GCSAM (GCET2, Abcam, clone EPR14333, 1:500) for 60 minutes at 36 °C. Staining was then visualized using OptiView DAB kit (Ventana Medical Systems). The immunohistochemistry was optimized on the control tissue with a known protein expression. We stained 6 cases, and two representative cases are shown.

### Statistics and reproducibility
Statistical analyses were performed using R v4.1.2 software. All $p$ values were calculated by two-sided analysis and considered statistically significant at $p < 0.05$. Student's $t$ test was used for comparisons of expression and DNA methylation levels. Fisher's exact test was used for group comparisons. Correlations across gene groups were also assessed by Fisher's exact test and corrected by employing the Benjamini-Hochberg method. For survival analysis, survival was estimated using the Kaplan–Meier method, and the log-rank test was used to assess differences between groups. For multivariate analysis, a Cox proportional hazards regression model was used to identify the risk factors associated with the OS rate. The model included *PHF6*, *RUNX1*, *U2AF1*, *ASXL1*, sex, and age (older than 60 years). Statistical analysis for the survival analysis was performed using the R package survival version 3.5-3. For differential expression gene analysis, we used the Bayesian method by the linear models for microarray expression data (limma) package version 3.50.0 in the R software[39]. To perform the enrichment of the difference between $PHF6^{MT}$ or $RUNX1^{MT}$ and WT AML, we used the GSEA software (v.4.3.2).

### Reporting summary
Further information on research design is available in the Nature Portfolio Reporting Summary linked to this article.

## Data availability
In 1465 cases from CCF, the raw data of whole exome or targeted capture sequencing in 401 cases are available in the dbGaP under accession codes, phs001898.v1.p1 [https://www.ncbi.nlm.nih.gov/projects/gap/cgi-bin/study.cgi?study_id=phs001898.v1.p1] or phs003303.v1.p1 [https://www.ncbi.nlm.nih.gov/projects/gap/cgi-bin/study.cgi?study_id=phs003303.v1.p1]. The remaining 1065 cases are derived under a CCF IRB-approved registry protocol to collect clinical data such as mutational status obtained from the clinical charts, which after data extraction, deidentified. IRB deemed this registry to be of minimal/less than minimal risk and thereby not requiring individual consent. Consequently, BAM data derived from clinical targeted capture sequencing stored by the pathology department at Cleveland Clinic were not available and are not obtainable under this protocol. However, complete annotated results (full mutational status) on per per-patient basis will be made available upon request by contacting the corresponding author: maciejj@ccf.org. The MLL data were from a

private company (MLL) that controls data release. The data, including the raw data from MLL cohort, is available upon request to the Munich Leukemia Institute (torsten.haferlach@mll.com). The processed 5109 data for whole genome sequencing is publicly available and can be found at [https://github.com/ardadurmaz/mds_latent and https://github.com/ardadurmaz/aml]. The 1869 data of whole-exome sequencing in public series (the Beat AML and the German-Austrian AML study cohorts) are available in the respective published articles[24,25]. All other sequencing information is provided in the Supplementary Information/ Data/ Tables. Molecular annotation is provided in the Supplemental Data of the submitted manuscript. Supplementary Data 1, 2, and 3 contain a list of patients with detailed annotation of karyotype, mutational status, and proteomics data, respectively. RNA expression and DNA methylation data in the Beat AML cohort are available online[24,26]. ChIPseq and the mass spec data were deposited in GEO (accession number, GSE229948), the BioProject (accession number, PRJNA956877), and the ProteomeXchange consortium (accession number, PXD042441), respectively. Access can be granted through dbGAP, GEO, and the ProteomeXchange contact can be made to Jaroslaw P. Maciejewski (maciejj@ccf.org). Because human genomic, phenotypic, or proteomic data is potentially sensitive information, the data must be shared in a manner consistent with the informed consent of the research participants, and the confidentiality of the data and the privacy of participants must be protected. Access to deposited human genomic data will be provided to research investigators who have certified their agreement with their institutions that they agree to the expectations and access conditions detailed in the dbGaP [https://sharing.nih.gov/accessing-data/accessing-genomic-data/how-to-request-and-access-datasets-from-dbgap]. There are no restrictions on who will be granted access. Source data are provided in this paper.

## Code availability

The scripts used for all analyses are deposited to https://github.com/ysokbt/PHF6 [https://github.com/ysokbt/PHF6] and are publicly available (https://doi.org/10.5281/zenodo.10380939)[43] [https://doi.org/10.5281/zenodo.10380939].

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

## Acknowledgements

We thank our sources of funding: grants R35HL135795 (to J.P.M.), The Leukemia & Lymphoma Society TRP Award 6645-22 (to J.P.M.), and the Edward P. Evans Foundation (to C.G.). We also thank Torsten Haferlach Leukämiediagnostik Stiftung for supporting this work. The authors also thank The Cancer Genome Atlas, The Beat AML Master Trial, and The German-Austrian Study Group for data accessibility. We are grateful to Ishani Nautiyal, Megan Reynold, and Michaela Witt for their excellent technical assistance.

## Author contributions

Authorship contributions are as follows: Y.K. designed the study, analyzed and interpreted the data, and wrote the manuscript; L.T., N.K., M.A. analyzed sequencing data and participated in the original data collection; W.B., T.K., B.P., H.A., M.Mori collected clinical data and edited the manuscript; J.B., H.J.R. performed immunohistochemistry; X.G., N.W., B.K.J. analyzed proteomics data; B.P.P. performed ChIPseq; C.G., A.D., V.V. took part in data interpretation and provided important feedback to the manuscript; M.Meggendorfer, T.H. shared sequencing data and edited the manuscript; J.P.M. conceived the study, interpreted the data and wrote the manuscript. All authors participated in discussions and interpretation of the data and results.

## Competing interests

This work was supported by an R35HL135795 (to J.P.M.), The Leukemia & Lymphoma Society TRP Award 6645-22 (to J.P.M.), and by a grant from the Edward P. Evans Foundation (to C.G.). The remaining authors declare no competing interests.
