## [Peer Review File · Nature Communications]

Molecular and clinical analyses of *PHF6* mutant myeloid neoplasia provide their pathogenesis and therapeutic targetingREVIEWER COMMENTS

Reviewer #1 (Remarks to the Author):

In this manuscript, a multi-omics approach was used to decipher the molecular and clinical properties of PHF6, a chromatin remodeling factor and a putative tumor suppressor gene in myeloid neoplasia (MN). Previous studies have shown that mutations of this gene (PHF6MT) are early and driver events in lymphoblastic leukemias. However, PHF6MT appears to be a secondary event in MN. The authors conducted what appears to be the largest study of this gene in MN. They found that PHF6 and RUNX1 are co-mutated in male patients. PHF6MT predicts unfavorable outcomes both independently and additively with mutations of RUNX1. In addition, the authors found that PHF6 physically interacts with RUNX1, a pioneer factor. This result, along with some others, led the authors to propose a model that the two genes function in the same pathway, but their sequential mutations are both permissive and additive in promoting MN by differentially regulating lymphocyte and myeloid genes. A weakness is that most results are correlative, and the proposed model is somewhat speculative. More studies on the causal effects of mutations would increase the impact of the work.

Other comments/suggestions:

1. The interaction in Fig. 4 is potentially important. However, it was only demonstrated between wild-type proteins. What might be the consequences of RUNX1 and/or PHF6 mutations on this interaction? Can the authors show that this interaction is maintained or disrupted by prognostically important mutations of PHF6 and RUNX1 (e.g., those co-mutated in Fig. 3D-F)? Similarly, do mutations of PHF6 affect its chromatin occupancy or that of RUNX1 and vice versa?
2. The authors proposed that PHF6 and RUNX1 co-occupy enhancers. However, the current results cannot exclude the possibility that these two proteins bind to the same loci but do so independently. It would be better if the authors can use Re-ChIP to prove, at least for some candidate genes, that PHF6 and RUNX1 are in the same complex on chromatin.
3. In Figs. 4B and 4C, there were several bands in the blots. Can the authors clearly label which band(s) belong to RUNX1 (Runx1) and PHF6 (Phf6)?
4. What are the datasets analyzed in Fig. 4F and/or 4G? The methods suggest that these are some microarray datasets, and the figure legends are not very helpful.
5. In addition to overall survivals, can the authors analyze the effect on event-free or relapse-free survivals that better correlate with disease progression?
6. Can the authors provide more details about the mutations of PHF6 and RUNX1 in patients showing additive negative outcomes (Figs. 3D-F)?
7. The authors have shown that RUNX1 is one of the most mutated genes in patients with PHF6MT. What about the frequency of PHF6 mutations in patients with RUNX1 mutations? This may provide further evidence that these two genes interact in the development of MN.

Reviewer #2 (Remarks to the Author):

The study by Kubota et al systematically evaluates the spectrum of PHF6 mutations across a large, multi-institutional cohort of 8443 myeloid neoplasms. This is an interesting study, and potential impactful for the field, but there are a number of technical and experimental concerns.

1. This is primarily a genomics paper yet none of the genomics data is publicly available, thus making it nearly impossible for other investigators to confirm these results. Data should be deposited for data sharing according to NIH standard (especially for those generated by CCF and supported by an R35 award). A statement that it will be available upon request is not appropriate.
2. There is no mention in the methods or the data availability statement about the transcriptome data used in Figure 4F. It'll be important to understand the tumor purity of these studies as this could account for the enrichment of lymphoid markers in these studies.
3. VAF/tumor fraction should be incorporated into the data presented in Figure 1. Absolute numbers should also be shown for Figure 1 panels B-D. Many of these events are likely rare in

these subsets.

4. Statistical review is recommended, especially regarding the lack of adjusted for multiple comparisons in Figure 1. Likewise, a thorough statistical review of the outcome data is likely warranted. Notably, a median followup of 13 months is likely too short for OS evaluation.

5. The proteomics data shows that SWI/SNF members seem to be the most significant interactors yet the focus is on RUNX1. Further, the present proteomics studies are intriguing but are overall incomplete. Do the PHF6 mutations observed in patients abolish these interactions? The selection of mouse splenocytes, which will be enriched in B and T cells, is odd for an endogenous validation of an interaction proposed to be important in myeloid cells. What about human CD34 cells? Is there any confirmation of these interactions in primary AML cells, with and without PHF6 mutations?

6. The isolated IHC and flow images in Figure 4 only add confusion. The IHC images lack sufficient resolution, and controls, to be meaningful. TdT-positive AMLs can occur outside of this context (e.g. PHF6 mutations) so the overall significance is not clear.

RESPONSE TO REVIEWERS' COMMENTS

We have extensively revised the manuscript to incorporate responses to the reviewers' critiques, which all were addressed including addition of new or improvement of existing figures. All changes have been indicated *in red font* in the manuscript and supplemental material and a point-by-point response to the comments is provided *in blue font* below.

Reviewer #1

Q1. The interaction in Fig. 4 is potentially important. However, it was only demonstrated between wild-type proteins. What might be the consequences of RUNX1 and/or PHF6 mutations on this interaction? Can the authors show that this interaction is maintained or disrupted by prognostically important mutations of PHF6 and RUNX1 (e.g., those co-mutated in Fig. 3D-F)? Similarly, do mutations of PHF6 affect its chromatin occupancy or that of RUNX1 and vice versa?

Authors' response: *We appreciate the reviewer's scholarly understanding of the importance of investigating the effect of PHF6 mutations on RUNX1 and the overall impact on the interactome. We focused on PHF6 centric prospective. We observed that the genetic alterations in PHF6 predominantly involved stopgains, frameshifts, and splice site mutations resulting in nonsense-mediated decay or produce abnormal proteins due to truncation of abnormal splicing (Supplementary Figure 13).*

Supplementary Figure 13

The fraction of truncating and non-truncating mutations in PHF6-mutated, RUNX1-mutated, and both-mutated samples. Samples with mutations in both genes have at least one truncating mutation in either **PHF6** or **RUNX1**.

This indicates that PHF6 interaction with RUNX1 would be lost in males with PHF6 mutations and in females

with *PHF6* mutations and chromosome X deletions. Western blotting for AML samples with frameshift or stopgain mutations in *PHF6* showed low expression reiterating the above described conclusion (Supplementary Figure 14).

Supplementary Figure 14

Western blots of PHF6 IP of nuclear protein extracts in AML samples with PHF6 mutations or wild-types. PHF6 bands are indicated by arrow in red color.

RUNX1-mutated samples with Runt domain nonsynonymous mutations accounted for 98% of nonsynonymous *RUNX1* mutations in our cohort. IP-Western analysis demonstrated that Runt domain mutation resulted in a less abundant co-immunoprecipitation of PHF6 than samples with wild-type *RUNX1* (Supplementary Figure 15).

Supplementary Figure 15

Western blots of endogenous PHF6 IP of nuclear protein extracts in AML samples with wild-type *RUNX1* or Runt domain mutations. 5% input, IgG control IP, and PHF6 IP product were run side by side. The same membrane was probed with an anti-rabbit monoclonal antibody to PHF6 and an antimouse monoclonal antibody to *RUNX1*. PHF6 and *RUNX1* bands are indicated by arrows in red color. The other *RUNX1* isoforms bands are indicated by arrows in blue color.

This result suggested that *RUNX1* interacts with PHF6 via Runt domain also fitting with the conclusion that

PHF6 mutant samples have no or weaker interaction between PHF6 and RUNX1. The fact that most of the *PHF6* mutated cases are present in cases with RUNX1 truncated after Runt domain also support the notion that the interaction between RUNX1 and PHF6 involves Runt domain (supplementary Figure 16). Of note is that chromatin occupancy of these mutated genes may be also affected because ePHD2 in PHF6 and Runt domain in RUNX1 responsible for chromatin binding are most often affected in nonsynonymous mutations in these genes. Consequently, the precise clarification of the interaction principles between wild-type and mutant RUNX1/PHF6 may require a separate analysis involving individual mutant types/locations of RUNX1, DNA binding, and analysis of the corepressor interaction and assessment of the wild-type allele effects.

We included now a figure to that end (Supplementary Figure 13-16) and comment about these important issues:

Supplementary Figure 16

Lollipop plot showing *RUNX1* stopgain mutations in male AML samples with double (*RUNX1* and *PHF6*) mutations, or only *RUNX1* mutations. Mutational groups are shown by colors as indicated. The numbers in circles indicate the number of cases.

Line 105-111: *“The ePHD2 domain would be essential for PHF6 function because 88% of nonsynonymous or in-frame mutations concentrated in the ePHD2 domain. This is in accordance with a previous study (PMID 32735658). As the ePHD2 domain is responsible for chromatin binding through recognition of acetylated histones, mutations in ePHD2 domain causing loss of ePHD2 domain would affect chromatin occupancy with PHF6. To support our hypothesis, we have also found that more than 90% of stopgain, frameshift, and splice site mutations are located anteriorly in the ePHD2 domain.”*

Line 252-267: *“In this study, we revealed the interaction between wild-type PHF6 and RUNX1. In case of presence of PHF6^{MT}, such interaction would be lost. Because, in our investigation, we observed that the genetic alterations in PHF6 predominantly involved stopgains, frameshifts, and splice site mutations resulting in nonsense-mediated decay or produce abnormal proteins due to truncation of abnormal splicing (Supplementary Figure 13). This indicates that PHF6 interaction with RUNX1 would be lost in males with PHF6^{MT} and in females with PHF6^{MT} and delX. Western blotting for AML samples with frameshift or stopgain mutations in PHF6 showed low expression reiterating the above describe conclusion (Supplementary Figure 14). RUNX1-mutated samples with Runt domain nonsynonymous mutations accounted for 98% of nonsynonymous RUNX1^{MT} in our cohort. IP-Western analysis demonstrated that Runt domain mutation resulted in a less abundant co-immunoprecipitation of PHF6 than samples with wild-type RUNX1 (Supplementary Figure 15). This result suggested that RUNX1 interacts with PHF6 via Runt domain also fitting with the conclusion that PHF6 mutant samples have no or weaker interaction between PHF6 and RUNX1. The fact that most of the PHF6^{MT} cases are present in cases with RUNX1 truncated after Runt domain also support the notion that the interaction between RUNX1 and PHF6 involves Runt domain (supplementary*

Figure 16).”

Line 301-311: *“By examining the interaction between WT PHF6 and RUNX1, we aimed to establish a baseline understanding of their functional relationship and the mechanistic implications of their interaction in cellular processes. We believe that comprehending the WT interaction is foundational for future investigations to evaluate the consequences of mutations in both PHF6 and RUNX1 on their interaction dynamics, as well as the potential disruption of chromatin occupancy. However, our study has several limitations. First, we were not able to show which domain in PHF6 is important for the interaction with RUNX1 or chromatin. As shown through our mutational analysis, we speculated that ePHD2 domain is essential for PHF6 function. Consequently, the precise clarification of the interaction principles between WT and mutant PHF6-RUNX1 may require a separate analysis involving individual mutant types/locations of RUNX1, DNA binding, and analysis of the corepressor interaction and assessment of the WT allele effects.”*

Q2. The authors proposed that PHF6 and RUNX1 co-occupy enhancers. However, the current results cannot exclude the possibility that these two proteins bind to the same loci but do so independently. It would be better if the authors can use Re-ChIP to prove, at least for some candidate genes, that PHF6 and RUNX1 are in the same complex on chromatin.

Authors’ response: We thank the reviewer for the comment. As you pointed out, our results do not entirely clarify at which enhancer site PHF6 and RUNX1 proteins act together. However, the protein complex immunoprecipitated by PHF6 antibody contains RUNX1 and vice versa (as shown in Figure 4B and 4C) confirming the point that PHF6 does interact with RUNX1. Our ChIPseq result showed that PHF6 and RUNX1 bind at multiple regions including active enhancers. Though we were unable to resolve the precise regions where PHF6 and RUNX1 bind, but it is unlikely that PHF6 and RUNX1 proteins bound independently to all the regions occupied by these proteins (please refer to Figure 4D and 4E).

Line 214-217: *“Consistent with our hypothesis and co-immunoprecipitation experiments, ChIPseq showed the co-localization of PHF6 and RUNX1 in multiple regions including active enhancers (Figure 4D-E and Supplementary Table 4-5).”*

Line 311-314: *“Second, it remains to be precisely elucidated which genomic regions PHF6 co-occupy with RUNX1. However, our ChIPseq result showed PHF6 and RUNX1 bind at multiple regions including active enhancers. While possible for some sites, it is unlikely that PHF6 and RUNX1 proteins bind independently to all the regions occupied by these proteins.”*

Q3. In Figs. 4B and 4C, there were several bands in the blots. Can the authors clearly label which band(s) belong to RUNX1 (Runx1) and PHF6 (Phf6)?

Authors’ response: We appreciate your comment regarding the labeling of bands in Figures 4B and 4C. We apologize for any confusion caused by the lack of clear identification of the bands corresponding to RUNX1 and PHF6 proteins. In our revised manuscript, we have addressed this concern by providing clear and specific labels for each band, ensuring accurate interpretation of the data. Also, we have added the sentences in Figure legends as follows:

Line 658-661: *“(B) Western blots of endogenous PHF6 IP of THP-1 nuclear protein extracts. 5% input, PHF6*

IP, and IgG control IP product were run side by side. The same membrane was probed with an anti-rabbit monoclonal antibody to PHF6 and an anti-mouse monoclonal antibody to RUNX1. PHF6 and RUNX1 bands are indicated by arrows in pink color.”

Line 662-665: *“(C) Reciprocal IPs of endogenous Phf6 and Runx1 with mouse spleen and bone marrow protein extracts. 5% input, Runx1 IP, Phf6 IP, and IgG control IP product were run side by side. The same membrane was probed with rabbit monoclonal antibody to Phf6 and mouse monoclonal antibody to Runx1. Phf6 and Runx1 bands are indicated by pink arrows.”*

Q4. What are the datasets analyzed in Fig. 4F and/or 4G? The methods suggest that these are some microarray datasets, and the figure legends are not very helpful.

Authors' response: We apologize for the lack of clarity. We used the Beat AML cohort for RNA-expression and DNA methylation analyses. To make it clear, we have rephrased and added the sentences as follows:

Line 330-331: *“Also we used the Beat AML cohort for RNA-expression and DNA methylation analyses (PMID 30333627 and 33707228).”*

Line 440: *“For RNA-expression analysis, we used the Beat AML cohort.”*

Line 612-616: *“(F) Comparisons of mRNA expression levels in genes on X chromosome between male (n = 139) and female (n=113) cases in the Beat AML cohort. The P value was calculated using the two-tailed Student's t-test. The mean, 25th, and 75th percentiles are represented in the box plots by the midline and box edges, respectively. The whiskers extend to 1.5 times the interquartile range. Dots represent each outlier of expression.”*

Line 617-620: *“(G) Comparisons of DNA methylation levels in the promoter regions of genes mapping on X chromosome between male (n=103) and female (n=99) cases in the Beat AML cohort. The mean, 25th, and 75th percentiles are represented in the box plots by the midline and box edges, respectively. The whiskers extend to 1.5 times the interquartile range. Dots represent each outlier of DNA methylation.”*

Q5. In addition to overall survivals, can the authors analyze the effect on event-free or relapse-free survivals

that better correlate with disease progression?

Authors' response: We thank the Reviewer for this important comment. We have analyzed leukemia-free survival in 4889 cases with available information of leukemic relapse and excluded open data cases with no relapse data. In accordance with overall survival data, the *PHF6*-mutated and the double-mutated cases revealed worse event-free survival. A detailed analysis of the double-mutated cases showed that these cases did not relapse, but did not achieve remission. Thus, conventional therapies would be not effective for these cases. We have incorporated the new supplementary figure (Supplementary Figure 7 and 10) and sentences as follows:

Line 172-176: *“Using our AML cohorts with available leukemic relapse information, event free survival (EFS) was also shorter in *PHF6*^{MT} cases (10% vs. 32% at 3 years, respectively; $p=0.00087$; Supplementary Figure 7A). Similar to OS, only *PHF6*^{MT} male cases showed significantly shorter EFS (8.5% vs. 30% at 3 years, respectively; $p=0.0074$; Supplementary Figure 7B-C).”*

Line 190-194: *“In accordance with OS, the double-mutated cases revealed worse EFS ($p<0.0001$ in all; $p<0.0001$ in male; $p=0.004$ in females; Supplementary Figure 10A-C). A detailed analysis of the double-mutated cases showed that most of these cases did not relapse, but did not achieve remission. Thus, conventional therapies would be not effective for these double-mutated cases.”*

Supplementary Figure 7

Event free survival for AML cases with or without *PHF6* mutations in the CCF and MLL cohort.

Kaplan-Meier survival curves of event free survival for AML in all (A), male (B), and female (C) cases.

Supplementary Figure 10

Event free survival for AML cases with or without *PHF6* mutations in the CCF and MLL cohort.

Kaplan-Meier survival curves of event free survival for female AML cases with double mutations (*PHF6* and *RUNX1*), single mutations, and negative cases in all (A), male (B), and female (C) cases.

Q6. Can the authors provide more details about the mutations of *PHF6* and *RUNX1* in patients showing additive negative outcomes (Figs. 3D-F)?

Authors' response: We appreciate your helpful suggestion. Please see the attached figure for the rebuttal. Panel A and B shows lollipop plots of *PHF6* mutations in cases with double (*PHF6* and *RUNX1*) mutations or only *PHF6* mutations, respectively. Also, Panel C and D shows lollipop plots of *RUNX1* mutations in cases with double (*PHF6* and *RUNX1*) mutations or only *RUNX1* mutations, respectively. There was no difference of mutated domains or mutational types between double-mutated and single-mutated cases. It is similar to *RUNX1* mutations that concentrated in the Runt domain in both double-mutated and single-mutated cases. Panel E and F shows the cell fraction with mutated *PHF6* or *RUNX1* allele in cases with double or single mutation, respectively. As *PHF6* is located on X chromosome, we used the cell fraction instead of the variant allele frequency. The cell fraction of each mutated gene also looks similar. Thus, it has meaning that both genes are mutated. *RUNX1* is a master regulator for myeloid differentiation. In the 2016 WHO classification, *RUNX1* mutations is considered as adverse risk factor with poor prognosis. This is because the loss of *RUNX1* function would impair differentiation to myeloid/lymphoid lineage as shown in the expression analysis (Supplementary Figure 12). Also, *PHF6* has function related to differentiation to myeloid/lymphoid lineages (Figure 4F and 4G). Thus, *PHF6* and *RUNX1* regulate differentiation to myeloid/lymphoid lineages either independently or cooperatively by interaction. In addition, *PHF6* plays a tumor suppressive role through interacting with DNA repair proteins such as *BLM*, *MSH2*, or *FANCI* (Figure 4A). Therefore, double mutants would show more aggressive phenotype and worse prognosis because mutations in *PHF6* and *RUNX1* result in loss of not only their individual functions, but also their functions through interaction. The manuscript has

been revised as follows:

Supplementary Figure 17

Lollipop plot of *PHF6* mutations in case with *PHF6* and *RUNX1* mutations (A) and only *PHF6* mutations (B).

Lollipop plot of *RUNX1* mutations in cases with *PHF6* and *RUNX1* mutations (C) and only *RUNX1*

mutations (D). The cell fraction with mutated *PHF6* or *RUNX1* allele in cases with *PHF6* and *RUNX1*

mutations (E) and *PHF6* or *RUNX1* mutations (F).

Line 287-290: “Comparing double-mutant (*PHF6* and *RUNX1*) and single-mutant (*PHF6* or *RUNX1*) cases, there was no difference of mutated domains or mutational types between double-mutated and single-mutated cases (Supplementary Figure 17A-D). It is similar to *RUNX1* mutations that concentrated in the Runt domain

in both double-mutated and single-mutated cases.”

Line 314-318: *“Finally, we were not able to show why PHF6 and RUNX1 mutations have additive effect. This would be due to the differentiation function of PHF6 and RUNX1, and tumor suppressive function of PHF6. To unveil this additive effect, we need to elucidate the functional consequences of these mutations on protein-protein interactions and downstream cellular processes.”*

Q7. The authors have shown that RUNX1 is one of the most mutated genes in patients with PHF6MT. What about the frequency of PHF6 mutations in patients with RUNX1 mutations? This may provide further evidence that these two genes interact in the development of MN.

Authors' response: We acknowledge that this is an important point. *PHF6* mutations co-occurred with *RUNX1* mutations in AML and MDS. In addition, this combination was more frequent only in male cases. In contrast, *PHF6* mutations were not associated with *RUNX1* mutations in MDS/MPN and MPN. The combination of *PHF6* and *RUNX1* may be not important for MDS/MPN and MPN or not permissive, though the number of cases is small. We have added the supplementary figure (Supplementary Figure 4) and sentences in Genomic landscape *PHF6*^{MT} of in Results section.

Line 142-146: *“PHF6*^{MT} *was significantly more prevalent among cases with RUNX1*^{MT} *compared to MN cases without RUNX1*^{MT} *(4.4 vs. 1.4%; p<0.001) and significantly co-occurred with RUNX1*^{MT} *only in male cases (Supplementary Figure 4). In addition, PHF6*^{MT} *was significantly more frequent in cases with RUNX1*^{MT} *in AML and MDS (p<0.001 and p=0.032, respectively).”*

Supplementary Figure 4

Comparisons of frequencies of *PHF6* mutations with or without *RUNX1* mutations based on each MN disease

Reviewer #2 (Remarks to the Author):

Q1. This is primarily a genomics paper yet none of the genomics data is publicly available, thus making it nearly impossible for other investigators to confirm these results. Data should be deposited for data sharing according to NIH standard (especially for those generated by CCF and supported by an R35 award). A statement that it will be available upon request is not appropriate.

Authors' response: We thank the reviewer for the comment. We have deposited our sequencing data in the dbGaP. Data is accessible through accession number phs001898.v1.p1 and phs003303.v1.p1. The raw proteomics data is deposited in the ProteomeXchange consortium (accession number, PXD042441) and the raw data of ChIPseq is deposited in GEO (accession number, GSE229948). Also, we have added and rephrased the sentences in the Data Availability section as follows:

Line 475-485: *“All the data supporting these findings, including DNA sequencing, ChIPseq, and the mass spec data were deposited in the dbGaP (accession number, phs001898.v1.p1 and phs003303.v1.p1), GEO (accession number, GSE229948), and the ProteomeXchange consortium (accession number, PXD042441), respectively. Molecular annotation is provided in the supplemental tables of the submitted manuscript. Supplementary Tables 1, 3, and 4 contain a list of patients with detailed annotation of karyotype, mutational status, and proteomics data, respectively. Public series (the BEAT AML Master Trial and the German-Austrian cohorts) are available in the respective published articles. Part of the data were extracted from open-access sources: https://github.com/ardadurmaz/mds_latent; <https://github.com/ardadurmaz/aml>. Whole genome sequencing data can be requested from the Munich Leukemia laboratory (torsten.haferlach@mll.com). All relevant data are available upon request by contacted the corresponding [author: maciej@ccf.org](mailto:maciej@ccf.org).”*

Q2. There is no mention in the methods or the data availability statement about the transcriptome data used in Figure 4F. It'll be important to understand the tumor purity of these studies as this could account for the enrichment of lymphoid markers in these studies.

Authors' response: We apologize for the lack of clarity. We used Beat AML cohorts for the RNA-expression and DNA methylation analyses. To make it clear, we have rephrased and added the sentences as follows:

Line 330-331: *“Also we used the Beat AML cohort for RNA-expression and DNA methylation analyses (PMID 30333627 and 33707228).”*

Line 440: *“For RNA-expression analysis, we used the Beat AML cohort.”*

Q3. VAF/tumor fraction should be incorporated into the data presented in Figure 1. Absolute numbers should also be shown for Figure 1 panels B-D. Many of these events are likely rare in these subsets.

Authors' response: We thank the reviewer for the comment. We have incorporated the cell fraction with *PHF6* mutation in the new Figure 1B to 1D. Also, we have added the sentences as follows:

Line: 605-606: “(B) Comparisons of frequencies of PHF6 mutations based on patients’ age groups. Each dot in the upper panel represents the cell fraction with PHF6 mutation.”

Line: 607-608: “(C) Comparisons of frequencies of PHF6 mutations based on MN disease in each sex. Each dot in the upper panel represents the cell fraction with PHF6 mutation.”

Line: 609-610: “(D) Comparisons of frequencies of PHF6 mutations based on each MN subtype. Each dot in the upper panel represents the cell fraction with PHF6 mutation.”

Q4. Statistical review is recommended, especially regarding the lack of adjusted for multiple comparisons in Figure 1. Likewise, a thorough statistical review of the outcome data is likely warranted.

Authors’ response: Thank you for your kind comment. As pointed out, we have reviewed all the statistics in this paper. We have corrected P values by employing the Benjamini-Hochberg method in Figure 1F-1G.

Line: 624-626: “P values were calculated by using two-tailed Student’s t-test and adjusted by Benjamini-Hochberg correction. *FDR<0.05, **FDR < 0.01, ***FDR <0.001.”

Notably, a median followup of 13 months is likely too short for OS evaluation.

Authors’ response: As you pointed out, we may overestimate the survival probability due to a shorter median follow-up period. However, we would like to show the negative impact of PHF6 on survival rather than long-term survival in this study. In addition, the log-rank test revealed significant difference between PHF6-mutated and wild-type cases. Thus, we have corrected the Kaplan-Meier curves in shorter time periods (60 to 36 months as twice as median follow-up period) to avoid overestimation and added the sentences in the discussion section as follows:

Line: 294-295: “However, long-term survival may be overestimated because of the short median follow-up period.”

Q5. The proteomics data shows that SWI/SNF members seem to be the most significant interactors yet the focus is on RUNX1.

Authors' response: Thank you for your response and additional insights regarding the proteomics data and the focus of our study on RUNX1. We appreciate your valuable input, and we would like to address your points and provide further clarification. Indeed, the significant interactions observed between PHF6 and the SWI/SNF and NuRD complexes in the proteomics data (Figure 4A) align with previous publications in T-

ALL suggesting that PHF6 may play an important role as a component of these chromatin remodeling complexes in AML. This reinforces the potential involvement of PHF6 in chromatin remodeling and gene regulation processes. However, interaction with transcription factors including RUNX1, CBFβ, and SPI1 has not been reported in any previous study. Thus, in this study, we focused to investigate the novel interaction between PHF6 and the pioneer transcription factor RUNX1. To make it clear, we have rephrased and added the sentences as follows:

Line: 202-206: *“Interestingly, PHF6 was also associated with several proteins in DNA repair and mRNA splicing such as MSH2, BLM, SRRT, or DDX5. PHF6 plays a role in chromatin remodeling, replication, and DNA repair not only in T-ALL but also in AML. However, in contrast to T-ALL, hematopoietic lineage defining pioneer transcription factors including, RUNX1, CBFβ, and SPI1 were among the most noticeable and functionally relevant co-immunoprecipitated proteins.”*

Further, the present proteomics studies are intriguing but are overall incomplete. Do the PHF6 mutations observed in patients abolish these interactions?

Authors’ response: We observed that the genetic alterations in *PHF6* predominantly involved stopgains, frameshifts, and splice site mutations resulting in nonsense-mediated decay or produce abnormal proteins due to truncation of abnormal splicing (Supplementary Figure 13).

Supplementary Figure 13

The fraction of truncating and non-truncating mutations in PHF6-mutated, RUNX1-mutated, and both-mutated samples. Samples with mutations in both genes have at least one truncating mutation in either ***PHF6*** or ***RUNX1***.

This indicates that PHF6 interaction with RUNX1 would be lost in males with *PHF6* mutations and in females with *PHF6* mutations and chromosome X deletions. Western blotting for AML samples with frameshift or stopgain mutations in *PHF6* showed or low expression reiterating the above describe conclusion (Supplementary Figure 14).

Supplementary Figure 14

Western blots of PHF6 IP of nuclear protein extracts in AML samples with PHF6 mutations or wild-types. PHF6 bands are indicated by arrow in red color.

Therefore, the manuscript has been revised as follows:

Line 252-259: *“In this study, we revealed the interaction between wild-type PHF6 and RUNX1. In case of presence of PHF6^{MT}, such interaction would be lost. Because, in our investigation, we observed that the genetic alterations in PHF6 predominantly involved stopgains, frameshifts, and splice site mutations resulting in nonsense-mediated decay or produce abnormal proteins due to truncation of abnormal splicing (Supplementary Figure 13). This indicates that PHF6 interaction with RUNX1 would be lost in males with PHF6^{MT} s and in females with PHF6^{MT} and DelX. Western blotting for AML samples with frameshift or stopgain mutations in PHF6 showed low expression reiterating the above describe conclusion (Supplementary Figure 14).”*

The selection of mouse splenocytes, which will be enriched in B and T cells, is odd for an endogenous validation of an interaction proposed to be important in myeloid cells. What about human CD34 cells? Is there any confirmation of these interactions in primary AML cells, with and without PHF6 mutations?

Authors' response: It is indeed noteworthy that we have observed consistent PHF6 and RUNX1 protein-protein interactions not only in mouse spleen and bone marrow cells (Figure 4C) but also in AML cell lines, THP1 (Figure 4B), albeit both RUNX1 and PHF6 were wild type in all instances. These findings support the notion that the interaction between PHF6 and RUNX1 is present in various cell types and contexts, including both normal physiological conditions and disease states. The inclusion of AML cells (THP1) in our study adds relevance to the context of myeloid malignancies, as it demonstrates the persistence of the PHF6-RUNX1 interaction in a cell line representative of the disease. Additionally, the consistent observations across multiple cell types further strengthen the significance of the interaction and support its potential importance in physiological processes. Then, we rephrase sentences as follows:

Line 208-214: *“We confirmed PHF6 and RUNX1 interaction with each other in normal physiological condition with reciprocal IP western-blot using the THP-1, mouse spleens, and bone marrow cells (Figure 4B-C and Supplementary Figure 11A-B). These results support the notion that the interaction between PHF6 and RUNX1 is present in various cell types and contexts, including both normal physiological conditions and disease states. The inclusion of AML cells (THP-1) in our study adds relevance to the context of myeloid malignancies, as it demonstrates the persistence of the PHF6-RUNX1 interaction in a cell line representative*

of the disease.”

Q6. The isolated IHC and flow images in Figure 4 only add confusion. The IHC images lack sufficient resolution, and controls, to be meaningful. TdT-positive AMLs can occur outside of this context (e.g. PHF6 mutations) so the overall significance is not clear.

Authors' response: Thank you for your kind suggestion. We have corrected the IHC figures by adding sufficient resolution, removed the flow cytometry figure, and added the sentences as follows:

Line: 674-675: *“(H) Positive cytoplasmic immunohistochemistry staining of LY9 in blasts in 2 PHF6-mutated AML cases. Inlet shows positive control of LY9 (x500).”*

Line: 676-677: *“(I) Positive cytoplasmic immunohistochemistry staining of GCSAM in blasts in 2 PHF6-mutated AML cases. Inlet shows positive control of GCSAM (x500).”*

REVIEWER COMMENTS

Reviewer #1 (Remarks to the Author):

All of my previous questions have been answered or addressed by the authors in this revision, which has significantly improved. I have no further concerns.

Reviewer #2 (Remarks to the Author):

The authors have done a fantastic job with addressing the issues and updating this revised manuscript. I have no further issues.

Reviewer #3 (Remarks to the Author):

In this paper, Kubota et al. studied the impact of PHF6 mutations (PHF6MT) in various myeloid neoplasms (MN). Here are some questions.

- Lines 113-117. "While PHF6MT were significantly more common in male cases (2.3 vs. 0.8% of all cases with M/F ratio of 3.5; $p < 0.001$; Figure 1C) with AML, no significant sex predilection was found in cases with MDS, myeloproliferative neoplasm (MPN) and myelodysplastic/myeloproliferative neoplasm (MDS/MPN)."

This section is misleading. The difference between males and females is similar across diseases. The reason why only AML is significant is because the number of samples in AML is much higher due to its relatively higher disease prevalence. As a result, for other diseases like MDS, there simply isn't enough power given the limited number of samples to detect differences of similar magnitude. This needs to be acknowledged. The same is true for 1D. In these bar plots, it is important that the sample sizes are included in these plots. Actually, my recommendation would be just remove all the results and conclusion for non-AML cancer types. There is just not enough samples to reach a definitive conclusion.

- In Figure 1E, the difference for UBA1 gene appear to have the largest difference. Why it is not significant? This is clearly problematic.
- Figures 1E, 1F and 1G, what is the order of the genes shown?
- Figure 1G, is the difference between the males and females significant for any of these genes?
- Figure 2A. What is the annotation of the black bar?
- Figure 3C and 3F. from month 12 to 36, the number of at risk reduce from 8 to 7, to 6 for PHF6, but the curve remains the same. Why?
- Lines 215-217. And Figure 4D. "ChIPseq showed the co-localization of PHF6 and RUNX1 in multiple regions including active enhancers (Figure 4D-E and Supplementary Table 4-5)."

This is not convincing. The authors also need to show ChIP-seq data from input as comparison. It is possible the similar pattern for PHF6 and RUX1 are due to the DNA accessibility, not co-binding.

RESPONSE TO REVIEWERS' COMMENTS

Reviewer #1

All of my previous questions have been answered or addressed by the authors in this revision, which has significantly improved. I have no further concerns.

Authors' response: We really appreciate your suggestions.

Reviewer #2

The authors have done a fantastic job with addressing the issues and updating this revised manuscript. I have no further issues.

Authors' response: We thank the reviewer for the comments which improved our paper.

Reviewer #3

Q1. "While PHF6MT were significantly more common in male cases (2.3 vs. 0.8% of all cases with M/F ratio of 3.5; $p < 0.001$; Figure 1C) with AML, no significant sex predilection was found in cases with MDS, myeloproliferative neoplasm (MPN) and myelodysplastic/myeloproliferative neoplasm (MDS/MPN)."

This section is misleading. The difference between males and females is similar across diseases. The reason why only AML is significant is because the number of samples in AML is much higher due to its relatively higher disease prevalence. As a result, for other diseases like MDS, there simply isn't enough power given the limited number of samples to detect differences of similar magnitude. This needs to be acknowledged. The same is true for 1D. In these bar plots, it is important that the sample sizes are included in these plots. Actually, my recommendation would be just remove all the results and conclusion for non-AML cancer types. There is just not enough samples to reach a definitive conclusion.

Authors' response: We thank the Reviewer for this important comment. According to your suggestion, we moved the results of MDS, MDS/MPN, and MPN from Figure 1C and 1D to Supplementary Figure 2.

Supplementary Figure 2

Supplementary Figure 2

(A) Comparisons of frequencies of *PHF6* mutations based on MN disease in each sex. Each dot in the upper panel represents the cell fraction with *PHF6* mutation.

(B) Comparisons of frequencies of *PHF6* mutations based on each MN subtype. Each dot in the upper panel represents the cell fraction with *PHF6* mutation.

We have also edited the sentences to acknowledge the size issue as follows:

Line 113-119: “*PHF6^{MT}* were significantly more common in male cases with AML (2.3 vs. 0.8% of all cases with M/F ratio of 3.5; $p < 0.001$; Figure 1C). *PHF6* is located on X chromosome, we used the cell fraction instead of the variant allele frequency. Notably, sAML showed higher frequency of *PHF6^{MT}* than primary AML (pAML; 2.9% vs. 1.4%; $p < 0.001$; Figure 1D). For MDS, myelodysplastic/myeloproliferative neoplasm (MDS/MPN), and MPN (myeloproliferative neoplasm), the numbers of cases were too small to determine sex differences (Supplementary Figure 2).”

Line 608-609: “(C) Comparisons of frequencies of *PHF6^{MT}* in each sex of AML cases. Each dot in the upper panel represents the cell fraction with *PHF6^{MT}*.”

Line 610-611: “(D) Comparisons of frequencies of *PHF6^{MT}* in pAML and sAML cases. Each dot in the upper panel represents the cell fraction with *PHF6^{MT}*.”

Q2. In Figure 1E, the difference for UBA1 gene appear to have the largest difference. Why it is not significant? This is clearly problematic.

Authors’ response: Thank you for your comment. Because we analyzed cases from Cleveland Clinic rather than BeatAML for the mutation rate of *UBA1*, we did not show significance in Figure 1E. According to your suggestion, we added the significance in Figure 1E.

Q3. Figures 1E, 1F and 1G, what is the order of the genes shown?

Authors' response: We apologize for the lack of clarity. The genes in Figure 1E-G were ordered by mutation rate in male cases. We added the sentences as follows:

Line 612-613: *"The genes are ordered by mutation rate in male cases."*

Q4. Figure 1G, is the difference between the males and females significant for any of these genes?

Authors' response: We apologize for the lack of clarity. We have added significance in Figure 1G.

Q5. Figure 2A. What is the annotation of the black bar?

Authors' response: We apologize for the lack of clarity. We have changed the annotation for the black bar as "Positive" and the white bar as "Negative" in Figure 2A.

Figure 2

Q6. Figure 3C and 3F. from month 12 to 36, the number of at risk reduce from 8 to 7, to 6 for PHF6, but the curve remains the same. Why?

Authors' response: We apologize for the lack of clarity. It was difficult to see the Figure because the line of the curve was thick. However, as the number of at risk reduces, the curve actually decreases after 12 months. Then we edited the Figure 3C and 3F to see clearly.

Q7. Figure 4D. “ChIPseq showed the co-localization of PHF6 and RUNX1 in multiple regions including active enhancers (Figure 4D-E and Supplementary Table 4-5).”

This is not convincing. The authors also need to show ChIP-seq data from input as comparison. It is possible the similar pattern for PHF6 and RUNX1 are due to the DNA accessibility, not co-binding.

Authors’ response: Thank you for your kind comment. As you pointed out, we have added the IgG control in Figure 4D. Also, we added the Supplementary Table regarding the peaks in IgG control (Supplementary Table 6). The merged peaks between PHF6 and RUNX1 in Figure 4E had been detected with the IgG control using the Homer software.

We edited the sentences as follows:

Line 668-669: “(D) Normalized distribution of PHF6, RUNX1 ChIPseq, and IgG control intensities in chromosome 1 in THP-1.”

REVIEWERS' COMMENTS

Reviewer #3 (Remarks to the Author):

The authors have addressed most of my questions.

However, for Q2, I think the revised Figure 1E is still problematic since for UBA1 gene, there is only one star, but most of the others have 3 stars, despite showing smaller differences in % change. Apparently, the differences shown in the bar plots do not match the significance indicators. I suggest the authors show the complete results on all the statistical tests performed in Figure 1, including mean, std.dev, test statistics, p-values, so we can have a better understanding of the discrepancy shown.

RESPONSE TO REVIEWERS' COMMENTS

Reviewer #3

Q1. The authors have addressed most of my questions.

Authors' response: We really appreciate your suggestions.

However, for Q2, I think the revised Figure 1E is still problematic since for UBA1 gene, there is only one star, but most of the others have 3 stars, despite showing smaller differences in % change. Apparently, the differences shown in the bar plots do not match the significance indicators. I suggest the authors show the complete results on all the statistical tests performed in Figure 1, including mean, std.dev, test statistics, p-values, so we can have a better understanding of the discrepancy shown.

Authors' response: We thank the Reviewer for this important comment. According to your suggestion, we added the table for this issue (Supplementary Table 1) and the sentences in the Figure legend as follows:

Line 625-628: *“Comparisons of frequencies of mutations in genes on X chromosome. The genes are ordered by mutation rate in male cases. The frequencies of UBA1 and PIGA mutations were based on other our cohorts.”*